# Provable Subspace Clustering:
# When LRR meets SSC

**Yu-Xiang Wang**
School of Computer Science
Carnegie Mellon University
Pittsburgh, PA 15213 USA
yuxiangw@cs.cmu.edu

**Huan Xu**
Dept. of Mech. Engineering
National Univ. of Singapore
Singapore, 117576
mpexuh@nus.edu.sg

**Chenlei Leng**
Department of Statistics
University of Warwick
Coventry, CV4 7AL, UK
C.Leng@warwick.ac.uk

## Abstract

Sparse Subspace Clustering (SSC) and Low-Rank Representation (LRR) are both considered as the state-of-the-art methods for *subspace clustering*. The two methods are fundamentally similar in that both are convex optimizations exploiting the intuition of "Self-Expressiveness". The main difference is that SSC minimizes the vector $\ell_1$ norm of the representation matrix to induce sparsity while LRR minimizes nuclear norm (aka trace norm) to promote a low-rank structure. Because the representation matrix is often simultaneously sparse and low-rank, we propose a new algorithm, termed Low-Rank Sparse Subspace Clustering (LRSSC), by combining SSC and LRR, and develops theoretical guarantees of when the algorithm succeeds. The results reveal interesting insights into the strength and weakness of SSC and LRR and demonstrate how LRSSC can take the advantages of both methods in preserving the "Self-Expressiveness Property" and "Graph Connectivity" at the same time.

## 1    Introduction

We live in the *big data era* – a world where an overwhelming amount of data is generated and collected every day, such that it is becoming increasingly impossible to process data in its raw form, even though computers are getting exponentially faster over time. Hence, *compact representations* of data such as low-rank approximation (e.g., PCA [13], Matrix Completion [4]) and sparse representation [6] become crucial in understanding the data with minimal storage. The underlying assumption is that high-dimensional data often lie in a low-dimensional subspace [4]). Yet, when such data points are generated from different sources, they form a *union of subspaces*. Subspace Clustering deals with exactly this structure by clustering data points according to their underlying subspaces. Application include motion segmentation and face clustering in computer vision [16, 8], hybrid system identification in control [26, 2], community clustering in social networks [12], to name a few.

Numerous algorithms have been proposed to tackle the problem. Recent examples include GPCA [25], Spectral Curvature Clustering [5], Sparse Subspace Clustering (SSC) [7, 8], Low Rank Representation (LRR) [17, 16] and its noisy variant LRSC [9] (for a more exhaustive survey of subspace clustering algorithms, we refer readers to the excellent survey paper [24] and the references therein). Among these algorithms, LRR and SSC, based on minimizing the nuclear norm and $\ell_1$ norm of the representation matrix respectively, remain the top performers on the Hopkins155 motion segmentation benchmark dataset [23]. Moreover, they are among the few subspace clustering algorithms supported with theoretic guarantees: Both algorithms are known to succeed when the subspaces are independent [27, 16]. Later, [8] showed that subspace being disjoint is sufficient for SSC to succeed[1], and [22] further relaxed this condition to include some cases of overlapping

subspaces. Robustness of the two algorithms has been studied too. Liu et al. [18] showed that a variant of LRR works even in the presence of some arbitrarily large outliers, while Wang and Xu [29] provided both deterministic and randomized guarantees for SSC when data are noisy or corrupted.

Despite LRR and SSC's success, there are questions unanswered. LRR has never been shown to succeed other than under the very restrictive "independent subspace" assumption. SSC's solution is sometimes too sparse that the affinity graph of data from a single subspace may not be a connected body [19]. Moreover, as our experiment with Hopkins155 data shows, the instances where SSC fails are often different from those that LRR fails. Hence, a natural question is whether combining the two algorithms lead to a better method, in particular since the underlying representation matrix we want to recover is *both low-rank and sparse* simultaneously.

In this paper, we propose Low-Rank Sparse Subspace Clustering (LRSSC), which minimizes a weighted sum of nuclear norm and vector 1-norm of the representation matrix. We show theoretical guarantees for LRSSC that strengthen the results in [22]. The statement and proof also shed insight on why LRR requires independence assumption. Furthermore, the results imply that there is a fundamental trade-off between the interclass separation and the intra-class connectivity. Indeed, our experiment shows that LRSSC works well in cases where data distribution is skewed (graph connectivity becomes an issue for SSC) and subspaces are not independent (LRR gives poor separation). These insights would be useful when developing subspace clustering algorithms and applications. We remark that in the general regression setup, the simultaneous nuclear norm and 1-norm regularization has been studied before [21]. However, our focus is on the subspace clustering problem, and hence the results and analysis are completely different.

## 2 Problem Setup

**Notations:** We denote the data matrix by $X \in \mathbb{R}^{n \times N}$, where each column of $X$ (normalized to unit vector) belongs to a union of $L$ subspaces

$$\mathcal{S}_1 \cup \mathcal{S}_2 \cup ... \cup \mathcal{S}_L.$$

Each subspace $\ell$ contains $N_\ell$ data samples with $N_1 + N_2 + ... + N_L = N$. We observe the noisy data matrix $X$. Let $X^{(\ell)} \in \mathbb{R}^{n \times N_\ell}$ denote the selection (as a set and a matrix) of columns in $X$ that belong to $\mathcal{S}_\ell \subset \mathbb{R}^n$, which is an $d_\ell$-dimensional subspace. Without loss of generality, let $X = [X^{(1)}, X^{(2)}, ..., X^{(L)}]$ be ordered. In addition, we use $\| \cdot \|$ to represent Euclidean norm (for vectors) or spectral norm (for matrices) throughout the paper.

**Method:** We solve the following convex optimization problem

$$\textbf{LRSSC}: \qquad \min_C \ \|C\|_* + \lambda \|C\|_1 \quad s.t. \quad X = XC, \quad \text{diag}(C) = 0. \tag{1}$$

Spectral clustering techniques (e.g., [20]) are then applied on the affinity matrix $W = |C| + |C|^T$ where $C$ is the solution to (1) to obtain the final clustering and $|\cdot|$ is the elementwise absolute value.

**Criterion of success:** In the subspace clustering task, as opposed to compressive sensing or matrix completion, there is no "ground-truth" $C$ to compare the solution against. Instead, the algorithm succeeds if each sample is expressed as a linear combination of samples belonging to the *same subspace*, i.e., the output matrix $C$ are *block diagonal* (up to appropriate permutation) with each subspace cluster represented by a disjoint block. Formally, we have the following definition.

**Definition 1** (Self-Expressiveness Property (SEP))**.** *Given subspaces $\{\mathcal{S}_\ell\}_{\ell=1}^L$ and data points $X$ from these subspaces, we say a matrix $C$ obeys Self-Expressiveness Property, if the nonzero entries of each $c_i$ ($i^{th}$ column of $C$) corresponds to only those columns of $X$ sampled from the same subspace as $x_i$.*

Note that the solution obeying SEP alone does not imply the clustering is correct, since each block may not be fully connected. This is the so-called "graph connectivity" problem studied in [19]. On the other hand, failure to achieve SEP does not necessarily imply clustering error either, as the spectral clustering step may give a (sometimes perfect) solution even when there are connections between blocks. Nevertheless, SEP is the condition that verifies the design intuition of SSC and LRR. Notice that if $C$ obeys SEP and each block is connected, we immediately get the correct clustering.

# 3 Theoretic Guanratees

## 3.1 The Deterministic Setup

Before we state our theoretical results for the deterministic setup, we need to define a few quantities.

**Definition 2** (Normalized dual matrix set). *Let $\{\Lambda_1(X)\}$ be the set of optimal solutions to*

$$\max_{\Lambda_1,\Lambda_2,\Lambda_3} \langle X, \Lambda_1 \rangle \quad s.t. \quad \|\Lambda_2\|_\infty \leq \lambda, \ \|X^T\Lambda_1 - \Lambda_2 - \Lambda_3\| \leq 1, \ \mathrm{diag}^\perp(\Lambda_3) = 0,$$

*where $\|\cdot\|_\infty$ is the vector $\ell_\infty$ norm and $\mathrm{diag}^\perp$ selects all the off-diagonal entries. Let $\Lambda^* = [\nu_1^*, ..., \nu_N^*] \in \{\Lambda_1(X)\}$ obey $\nu_i^* \in \mathrm{span}(X)$ for every $i = 1, ..., N$.[2] For every $\Lambda = [\nu_1, ..., \nu_N] \in \{\Lambda_1(X)\}$, we define* normalized dual matrix $V$ for $X$ *as*

$$V(X) \triangleq \left[ \frac{\nu_1}{\|\nu_1^*\|}, ..., \frac{\nu_N}{\|\nu_N^*\|} \right],$$

*and the* normalized dual matrix set $\{V(X)\}$ *as the collection of $V(X)$ for all $\Lambda \in \{\Lambda_1(X)\}$.*

**Definition 3** (Minimax subspace incoherence property). *Compactly denote $V^{(\ell)} = V(X^{(\ell)})$. We say the vector set $X^{(\ell)}$ is $\mu$-incoherent to other points if*

$$\mu \geq \mu(X^{(\ell)}) := \min_{V^{(\ell)} \in \{V^{(\ell)}\}} \max_{x \in X \backslash X^{(\ell)}} \|{V^{(\ell)}}^T x\|_\infty.$$

The incoherence $\mu$ in the above definition measures how *separable* the sample points in $\mathcal{S}_\ell$ are against sample points in other subspaces (small $\mu$ represents more separable data). Our definition differs from Soltanokotabi and Candes's definition of subspace incoherence [22] in that it is defined as a minimax over all possible dual directions. It is easy to see that $\mu$-incoherence in [22, Definition 2.4] implies $\mu$-minimax-incoherence as their dual direction are contained in $\{V(X)\}$. In fact, in several interesting cases, $\mu$ can be significantly smaller under the new definition. We illustrate the point with the two examples below and leave detailed discussions in the supplementary materials.

**Example 1** (Independent Subspace). Suppose the subspaces are independent, i.e., $\dim(\mathcal{S}_1 \oplus ... \oplus \mathcal{S}_L) = \sum_{\ell=1,...,L} \dim(\mathcal{S}_\ell)$, then all $X^{(\ell)}$ are 0-incoherent under our Definition 3. This is because for each $X^{(\ell)}$ one can always find a dual matrix $V^{(\ell)} \in \{V^{(\ell)}\}$ whose column space is orthogonal to the span of all other subspaces. To contrast, the incoherence parameter according to Definition 2.4 in [22] will be a positive value, potentially large if the angles between subspaces are small.

**Example 2** (Random except 1 subspace). Suppose we have $L$ disjoint 1-dimensional subspaces in $\mathbb{R}^n$ ($L > n$). $\mathcal{S}_1, ..., \mathcal{S}_{L-1}$ subspaces are randomly drawn. $\mathcal{S}_L$ is chosen such that its angle to one of the $L - 1$ subspace, say $\mathcal{S}_1$, is $\pi/6$. Then the incoherence parameter $\mu(X^{(L)})$ defined in [22] is at least $\cos(\pi/6)$. However under our new definition, it is not difficult to show that $\mu(X^{(L)}) \leq 2\sqrt{\frac{6 \log(L)}{n}}$ with high probability[3].

The result also depends on the smallest singular value of a rank-$d$ matrix (denoted by $\sigma_d$) and the inradius of a convex body as defined below.

**Definition 4** (inradius). *The inradius of a convex body $\mathcal{P}$, denoted by $r(\mathcal{P})$, is defined as the radius of the largest Euclidean ball inscribed in $\mathcal{P}$.*

The smallest singular value and inradius measure how *well-represented* each subspace is by its data samples. Small inradius/singular value implies either insufficient data, or skewed data distribution, in other word, it means that the subspace is "*poorly represented*". Now we may state our main result.

**Theorem 1** (LRSSC). *Self-expressiveness property holds for the solution of* (1) *on the data $X$ if there exists a weighting parameter $\lambda$ such that for all $\ell = 1, ..., L$, one of the following two conditions holds:*

$$\mu(X^{(\ell)})(1 + \lambda\sqrt{N_\ell}) < \lambda \min_k \sigma_{d_\ell}(X_{-k}^{(\ell)}), \tag{2}$$

$$\text{or} \quad \mu(X^{(\ell)})(1 + \lambda) < \lambda \min_k r(\mathrm{conv}(\pm X_{-k}^{(\ell)})), \tag{3}$$

where $X_{-k}$ denotes $X$ with its $k^{th}$ column removed and $\sigma_{d_\ell}(X_{-k}^{(\ell)})$ represents the $d_\ell^{th}$ (smallest non-zero) singular value of the matrix $X_{-k}^{(\ell)}$.

We briefly explain the intuition of the proof. The theorem is proven by duality. First we write out the dual problem of (1),

**Dual LRSSC** : $\displaystyle\max_{\Lambda_1,\Lambda_2,\Lambda_3} \langle X, \Lambda_1 \rangle$ s.t. $\|\Lambda_2\|_\infty \leq \lambda$, $\|X^T\Lambda_1 - \Lambda_2 - \Lambda_3\| \leq 1$, $\mathrm{diag}^\perp(\Lambda_3) = 0$.

This leads to a set of optimality conditions, and leaves us to show the existence of a dual certificate satisfying these conditions. We then construct two levels of fictitious optimizations (which is the main novelty of the proof) and *construct a dual certificate from the dual solution of the fictitious optimization problems*. Under condition (2) and (3), we establish this dual certifacte meets all optimality conditions, hence certifying that SEP holds. Due to space constraints, we defer the detailed proof to the supplementary materials and focus on the discussions of the results in the main text.

**Remark 1** (SSC). Theorem 1 can be considered a generalization of Theorem 2.5 of [22]. Indeed, when $\lambda \to \infty$, (3) reduces to the following

$$\mu(X^{(\ell)}) < \min_k r(\mathrm{conv}(\pm X_{-k}^{(\ell)})).$$

The readers may observe that this is exactly the same as Theorem 2.5 of [22], with the only difference being the definition of $\mu$. Since our definition of $\mu(X^{(\ell)})$ is tighter (i.e., smaller) than that in [22], our guarantee for SSC is indeed stronger. Theorem 1 also implies that the good properties of SSC (such as overlapping subspaces, large dimension) shown in [22] are also valid for LRSSC for a range of $\lambda$ greater than a threshold.

To further illustrate the key difference from [22], we describe the following scenario.

**Example 3** (Correlated/Poorly Represented Subspaces). Suppose the subspaces are poorly represented, i.e., the inradius $r$ is small. If furthermore, the subspaces are highly correlated, i.e., canonical angles between subspaces are small, then the subspace incoherence $\mu'$ defined in [22] can be quite large (close to 1). Thus, the succeed condition $\mu' < r$ presented in [22] is violated. This is an important scenario because real data such as those in Hopkins155 and Extended YaleB often suffer from both problems, as illustrated in [8, Figure 9 & 10]. Using our new definition of incoherence $\mu$, as long as the subspaces are "sufficiently independent"[4] (regardless of their correlation) $\mu$ will assume very small values (e.g., Example 2), making SEP possible even if $r$ is small, namely when subspaces are poorly represented.

**Remark 2** (LRR). The guarantee is the strongest when $\lambda \to \infty$ and becomes superficial when $\lambda \to 0$ unless subspaces are independent (see Example 1). This seems to imply that the "independent subspace" assumption used in [16, 18] to establish sufficient conditions for LRR (and variants) to work is unavoidable.[5] On the other hand, for each problem instance, there is a $\lambda^*$ such that whenever $\lambda > \lambda^*$, the result satisfies SEP, so we should expect phase transition phenomenon when tuning $\lambda$.

**Remark 3** (A tractable condition). Condition (2) is based on singular values, hence is computationally tractable. In contrast, the verification of (3) or the deterministic condition in [22] is NP-Complete, as it involves computing the inradii of $\mathcal{V}$-Polytopes [10]. When $\lambda \to \infty$, Theorem 1 reduces to the first computationally tractable guarantee for SSC that works for disjoint and potentially overlapping subspaces.

## 3.2 Randomized Results

We now present results for the random design case, i.e., data are generated under some random models.

**Definition 5** (Random data). *"**Random sampling**" assumes that for each $\ell$, data points in $X^{(\ell)}$ are iid uniformly distributed on the unit sphere of $\mathcal{S}_\ell$. "**Random subspace**" assumes each $\mathcal{S}_\ell$ is generated independently by spanning $d_\ell$ iid uniformly distributed vectors on the unit sphere of $\mathbb{R}^n$.*

**Lemma 1** (Singular value bound)**.** *Assume* random sampling. *If $d_\ell < N_\ell < n$, then there exists an absolute constant $C_1$ such that with probability of at least $1 - N_\ell^{-10}$,*

$$\sigma_{d_\ell}(X) \geq \frac{1}{2}\left(\sqrt{\frac{N_\ell}{d_\ell}} - 3 - C_1\sqrt{\frac{\log N_\ell}{d_\ell}}\right), \qquad or \ simply \qquad \sigma_{d_\ell}(X) \geq \frac{1}{4}\sqrt{\frac{N_\ell}{d_\ell}},$$

*if we assume $N_\ell \geq C_2 d_\ell$, for some constant $C_2$.*

**Lemma 2** (Inradius bound [1, 22])**.** *Assume* random sampling *of $N_\ell = \kappa_\ell d_\ell$ data points in each $\mathcal{S}_\ell$, then with probability larger than $1 - \sum_{\ell=1}^{L} N_\ell e^{-\sqrt{d_\ell N_\ell}}$*

$$r(\mathrm{conv}(\pm X_{-k}^{(\ell)})) \geq c(\kappa_\ell)\sqrt{\frac{\log(\kappa_\ell)}{2d_\ell}} \ for \ all \ pairs \ (\ell, k).$$

*Here, $c(\kappa_\ell)$ is a constant depending on $\kappa_\ell$. When $\kappa_\ell$ is sufficiently large, we can take $c(\kappa_\ell) = 1/\sqrt{8}$.*

Combining Lemma 1 and Lemma 2, we get the following remark showing that conditions (2) and (3) are complementary.

**Remark 4.** Under the *random sampling* assumption, when $\lambda$ is smaller than a threshold, the singular value condition (2) is better than the inradius condition (3). Specifically, $\sigma_{d_\ell}(X) > \frac{1}{4}\sqrt{\frac{N_\ell}{d_\ell}}$ with high probability, so for some constant $C > 1$, the singular value condition is strictly better if

$$\lambda < \frac{C\left(\sqrt{N_\ell} - \sqrt{\log(N_\ell/d_\ell)}\right)}{\sqrt{N_\ell}\left(1 + \sqrt{\log(N_\ell/d_\ell)}\right)}, \qquad or \ when \ N_\ell \ is \ large, \quad \lambda < \frac{C}{1 + \sqrt{\log(N_\ell/d_\ell)}}.$$

By further assuming *random subspace*, we provide an upper bound of the incoherence $\mu$.

**Lemma 3** (Subspace incoherence bound)**.** *Assume* random subspace *and* random sampling. *It holds with probability greater than $1 - 2/N$ that for all $\ell$,*

$$\mu(X^{(\ell)}) \leq \sqrt{\frac{6\log N}{n}}.$$

Combining Lemma 1 and Lemma 3, we have the following theorem.

**Theorem 2** (LRSSC for random data)**.** *Suppose $L$ rank-$d$ subspace are uniformly and independently generated from $\mathbb{R}^n$, and $N/L$ data points are uniformly and independently sampled from the unit sphere embedded in each subspace, furthermore $N > CdL$ for some absolute constant $C$, then SEP holds with probability larger than $1 - 2/N - 1/(Cd)^{10}$, if*

$$d < \frac{n}{96\log N}, \quad for \ all \quad \lambda > \frac{1}{\sqrt{\frac{N}{L}}\left(\sqrt{\frac{n}{96d\log N}} - 1\right)}. \tag{4}$$

The above condition is obtained from the singular value condition. Using the inradius guarantee, combined with Lemma 2 and 3, we have a different succeed condition requiring $d < \frac{n\log(\kappa)}{96\log N}$ for all $\lambda > \frac{1}{\sqrt{\frac{n\log\kappa}{96d\log N}} - 1}$. Ignoring constant terms, the condition on $d$ is slightly better than (4) by a *log* factor but the range of valid $\lambda$ is significantly reduced.

## 4 Graph Connectivity Problem

The graph connectivity problem concerns when SEP is satisfied, whether each block of the solution $C$ to LRSSC represents a connected graph. The graph connectivity problem concerns whether each disjoint block (since SEP holds true) of the solution $C$ to LRSSC represents a connected graph. This is equivalent to the connectivity of the solution of the following fictitious optimization problem, where each sample is constrained to be represented by the samples of the same subspace,

$$\min_{C^{(\ell)}} \|C^{(\ell)}\|_* + \lambda\|C^{(\ell)}\|_1 \quad s.t. \quad X^{(\ell)} = X^{(\ell)}C^{(\ell)}, \quad \mathrm{diag}(C^{(\ell)}) = 0. \tag{5}$$

The graph connectivity for SSC is studied by [19] under deterministic conditions (to make the problem well-posed). They show by a negative example that even if the well-posed condition is satisfied, the solution of SSC may not satisfy graph connectivity if the dimension of the subspace is greater than 3. On the other hand, graph connectivity problem is not an issue for LRR: as the following proposition suggests, the intra-class connections of LRR's solution are inherently dense (fully connected).

**Proposition 1.** *When the subspaces are independent, $X$ is not full-rank and the data points are randomly sampled from a unit sphere in each subspace, then the solution to LRR, i.e.,*

$$\min_{C} \|C\|_* \quad s.t. \quad X = XC,$$

*is class-wise dense, namely each diagonal block of the matrix $C$ is all non-zero.*

The proof makes use of the following lemma which states the closed-form solution of LRR.

**Lemma 4** ([16]). *Take skinny SVD of data matrix $X = U\Sigma V^T$. The closed-form solution to LRR is the shape interaction matrix $C = VV^T$.*

Proposition 1 then follows from the fact that each entry of $VV^T$ has a continuous distribution, hence the probability that any is exactly zero is negligible (a complete argument is given in the supplementary).

Readers may notice that when $\lambda \to 0$, (5) is not exactly LRR, but with an additional constraint that diagonal entries are zero. We suspect this constrained version also have dense solution. This is demonstrated numerically in Section 6.

# 5 Practical issues

## 5.1 Data noise/sparse corruptions/outliers

The natural extension of LRSSC to handle noise is

$$\min_{C} \frac{1}{2}\|X - XC\|_F^2 + \beta_1\|C\|_* + \beta_2\|C\|_1 \quad s.t. \quad \text{diag}(C) = 0. \tag{6}$$

We believe it is possible (but maybe tedious) to extend our guarantee to this noisy version following the strategy of [29] which analyzed the noisy version of SSC. This is left for future research.

According to the noisy analysis of SSC, a rule of thumb of choosing the scale of $\beta_1$ and $\beta_2$ is

$$\beta_1 = \frac{\sigma(\frac{1}{1+\lambda})}{\sqrt{2\log N}}, \qquad\qquad \beta_2 = \frac{\sigma(\frac{\lambda}{1+\lambda})}{\sqrt{2\log N}},$$

where $\lambda$ is the tradeoff parameter used in noiseless case (1), $\sigma$ is the estimated noise level and $N$ is the total number of entries.

In case of sparse corruption, one may use $\ell_1$ norm penalty instead of the Frobenious norm. For outliers, SSC is proven to be robust to them under mild assumptions [22], and we suspect a similar argument should hold for LRSSC too.

## 5.2 Fast Numerical Algorithm

As subspace clustering problem is usually large-scale, off-the-shelf SDP solvers are often too slow to use. Instead, we derive *alternating direction methods of multipliers* (ADMM) [3], known to be scalable, to solve the problem numerically. The algorithm involves separating out the two objectives and diagonal constraints with dummy variables $C_2$ and $J$ like

$$\min_{C_1,C_2,J} \|C_1\|_* + \lambda\|C_2\|_1$$
$$s.t. \quad X = XJ, \quad J = C_2 - \text{diag}(C_2), \quad J = C_1, \tag{7}$$

and update $J, C_1, C_2$ and the three dual variables alternatively. Thanks to the change of variables, all updates can be done in closed-form. To further speed up the convergence, we adopt the adaptive penalty mechanism of Lin et.al [15], which in some way ameliorates the problem of tuning numerical parameters in ADMM. Detailed derivations, update rules, convergence guarantee and the corresponding ADMM algorithm for the noisy version of LRSSC are made available in the supplementary materials.

# 6 Numerical Experiments

To verify our theoretical results and illustrate the advantages of LRSSC, we design several numerical experiments. Due to space constraints, we discuss only two of them in the paper and leave the rest to the supplementary materials. In all our numerical experiments, we use the ADMM implementation of LRSSC with fixed set of numerical parameters. The results are given against an exponential grid of $\lambda$ values, so comparisons to only 1-norm (SSC) and only nuclear norm (LRR) are clear from two ends of the plots.

## 6.1 Separation-Sparsity Tradeoff

We first illustrate the tradeoff of the solution between obeying SEP and being connected (this is measured using the intra-class sparsity of the solution). We randomly generate $L$ subspaces of dimension 10 from $\mathbb{R}^{50}$. Then, 50 unit length random samples are drawn from each subspace and we concatenate into a $50 \times 50L$ data matrix. We use Relative Violation [29] to measure of the violation of SEP and Gini Index [11] to measure the intra-class sparsity[6]. These quantities are defined below:

$$\text{RelViolation}\,(C, \mathcal{M}) = \frac{\sum_{(i,j) \notin \mathcal{M}} |C|_{i,j}}{\sum_{(i,j) \in \mathcal{M}} |C|_{i,j}},$$

where $\mathcal{M}$ is the index set that contains all $(i, j)$ such that $x_i, x_j \in S_\ell$ for some $\ell$.

GiniIndex $(C, \mathcal{M})$ is obtained by first sorting the absolute value of $C_{ij \in \mathcal{M}}$ into a non-decreasing sequence $\vec{c} = [c_1, ..., c_{|\mathcal{M}|}]$, then evaluate

$$\text{GiniIndex}\,(\text{vec}(C_{\mathcal{M}})) = 1 - 2 \sum_{k=1}^{|\mathcal{M}|} \frac{c_k}{\|\vec{c}\|_1} \left( \frac{|\mathcal{M}| - k + 1/2}{|\mathcal{M}|} \right).$$

Note that RelViolation takes the value of $[0, \infty]$ and SEP is attained when RelViolation is zero. Similarly, Gini index takes its value in $[0, 1]$ and it is larger when intra-class connections are sparser.

The results for $L = 6$ and $L = 11$ are shown in Figure 1. We observe phase transitions for both metrics. When $\lambda = 0$ (corresponding to LRR), the solution does not obey SEP even when the independence assumption is only slightly violated ($L = 6$). When $\lambda$ is greater than a threshold, RelViolation goes to zero. These observations match Theorems 1 and 2. On the other hand, when $\lambda$ is large, intra-class sparsity is high, indicating possible disconnection within the class.

Moreover, we observe that there exists a range of $\lambda$ where RelViolation reaches zero yet the sparsity level does not reaches its maximum. This justifies our claim that the solution of LRSSC, taking $\lambda$ within this range, can achieve SEP and at the same time keep the intra-class connections relatively dense. Indeed, for the subspace clustering task, a good tradeoff between separation and intra-class connection is important.

## 6.2 Skewed data distribution and model selection

In this experiment, we use the data for $L = 6$ and combine the first two subspaces into one 20-dimensional subspace and randomly sample 10 more points from the new subspace to "connect" the 100 points from the original two subspaces together. This is to simulate the situation when data distribution is skewed, i.e., the data samples within one subspace has two dominating directions. The skewed distribution creates trouble for model selection (judging the number of subspaces), and intuitively, the graph connectivity problem might occur.

We find that model selection heuristics such as the spectral gap [28] and spectral gap ratio [14] of the normalized Laplacian are good metrics to evaluate the quality of the solution of LRSSC. Here the correct number of subspaces is 5, so the spectral gap is the difference between the $6^{th}$ and $5^{th}$ smallest singular value and the spectral gap ratio is the ratio of adjacent spectral gaps. The larger these quantities, the better the affinity matrix reveals that the data contains 5 subspaces.

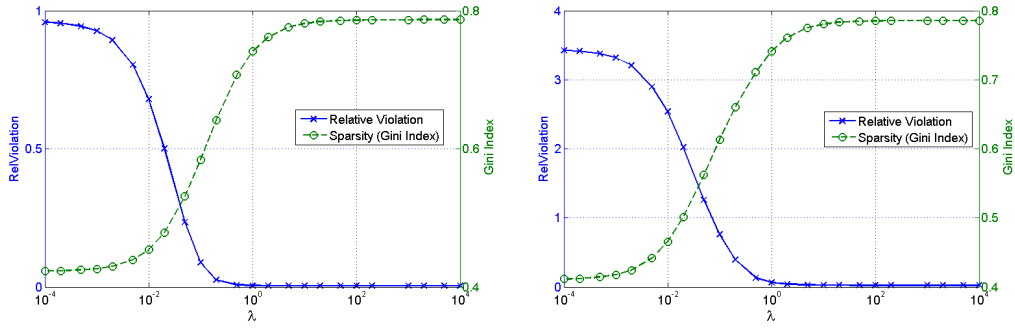

Figure 1: Illustration of the separation-sparsity trade-off. Left: 6 subspaces. Right: 11 subspace.

Figure 2 demonstrates how singular values change when $\lambda$ increases. When $\lambda = 0$ (corresponding to LRR), there is no significant drop from the $6^{th}$ to the $5^{th}$ singular value, hence it is impossible for either heuristic to identify the correct model. As $\lambda$ increases, the last 5 singular values gets smaller and become almost zero when $\lambda$ is large. Then the 5-subspace model can be correctly identified using spectral gap ratio. On the other hand, we note that the $6^{th}$ singular value also shrinks as $\lambda$ increases, which makes the spectral gap very small on the SSC side and leaves little robust margin for correct model selection against some violation of SEP. As is shown in Figure 3, the largest spectral gap and spectral gap ratio appear at around $\lambda = 0.1$, where the solution is able to benefit from both the better separation induced by the 1-norm factor and the relatively denser connections promoted by the nuclear norm factor.

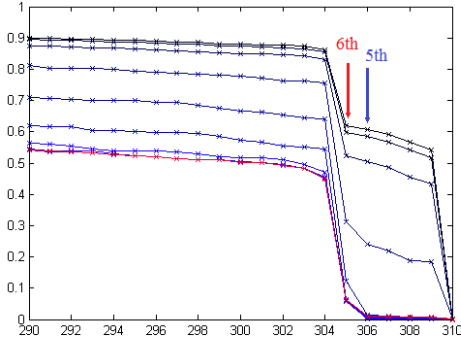

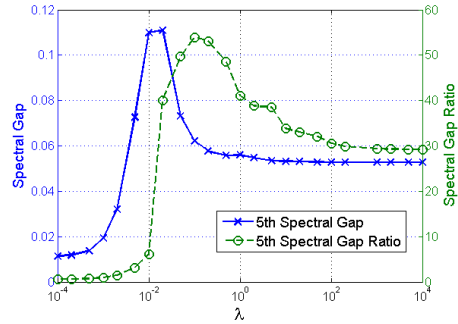

Figure 2: Last 20 singular values of the normalized Laplacian in the skewed data experiment.

Figure 3: Spectral Gap and Spectral Gap Ratio in the skewed data experiment.

## 7 Conclusion and future works

In this paper, we proposed LRSSC for the subspace clustering problem and provided theoretical analysis of the method. We demonstrated that LRSSC is able to achieve perfect SEP for a wider range of problems than previously known for SSC and meanwhile maintains denser intra-class connections than SSC (hence less likely to encounter the "graph connectivity" issue). Furthermore, the results offer new understandings to SSC and LRR themselves as well as problems such as skewed data distribution and model selection. An important future research question is to mathematically define the concept of the graph connectivity, and establish conditions that perfect SEP and connectivity indeed occur together for some non-empty range of $\lambda$ for LRSSC.

**Acknowledgments**

H. Xu is partially supported by the Ministry of Education of Singapore through AcRF Tier Two grant R-265-000-443-112 and NUS startup grant R-265-000-384-133.

## Footnotes

[1] Disjoint subspaces only intersect at the origin. It is a less restrictive assumption comparing to independent subspaces, e.g., 3 coplanar lines passing the origin are not independent, but disjoint.

[2]If this is not unique, pick the one with least Frobenious norm.

[3]The full proof is given in the supplementary. Also it is easy to generalize this example to $d$-dimensional subspaces and to "random except $K$ subspaces".

[4]Due to space constraint, the concept is formalized in supplementary materials.

[5]Our simulation in Section 6 also supports this conjecture.

[6]We choose Gini Index over the typical $\ell_0$ to measure sparsity as the latter is vulnerable to numerical inaccuracy.

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
