[Supplementary Material · LRSSC_nips_supplementary.pdf]

# Supplementary material for "Provable Subspace Clustering: When LRR meets SSC"

**Yu-Xiang Wang**
School of Computer Science
Carnegie Mellon University
Pittsburgh, PA 15213 USA
yuxiangw@cs.cmu.edu

**Huan Xu**
Dept. of Mech. Engineering
National Univ. of Singapore
Singapore, 117576
mpexuh@nus.edu.sg

**Chenlei Leng**
Department of Statistics
University of Warwick
Coventry, CV4 7AL, UK
C.Leng@warwick.ac.uk

The supplementary material is organized as follows. In Section A and B, we provide the detailed proof of respectively the deterministic and randomized guarantee for LRSSC. In Section C, we derive the fast Alternating Direction Methods of Multipliers (ADMM) algorithm for LRSSC and NoisyLRSSC and verify its convergence guarantee. In Section D, additional numerical experiments of LRSSC are provided, including the real data experiments on Hopkins155 motion segmentation dataset[16]. In Section E, we provide in-depth discussions on the Minimax Subspace Incoherence Property, illustrating its main difference to Subspace Incoherence Property in [14] with intuitive examples and proofs. Also in this section, we formalize the concept of "sufficiently independent" that appeared in Example 3 of the main paper. In Section F, the proof to some stand-alone claims in the paper are given, including the graph connectivity of LRR and the computational tractability of the singular value condition. Finally, for readers' easy reference, we attach a table of symbols and notations at the very end of the supplementary material.

## A  Proof of Theorem 1 (the deterministic result)

Theorem 1 is proven by duality. As described in the main text, it involves constructing two levels of fictitious optimizations. For convenience, we illustrate the proof with only three subspaces. Namely, $X = [X^{(1)} X^{(2)} X^{(3)}]$ and $\mathcal{S}_1 \, \mathcal{S}_2 \, \mathcal{S}_3$ are all $d$-dimensional subspaces. Having more than 3 subspaces and subspaces of different dimensions are perfectly fine and the proof will be the same.

### A.1  Optimality condition

We start by describing the subspace projection critical in the proof of matrix completion and RPCA[5, 4]. We need it to characterize the subgradient of nuclear norm.

Define projection $\mathcal{P}_T$ (and $\mathcal{P}_{T\perp}$) to both column and row space of low-rank matrix $C$ (and its complement) as

$$\mathcal{P}_T(X) = UU^T X + XVV^T - UU^T XVV^T,$$

$$\mathcal{P}_{T\perp}(X) = (I - UU^T)X(I - VV^T),$$

where $UU^T$ and $VV^T$ are projections matrix defined from skinny SVD of $C = U\Sigma V^T$.

**Lemma A.1.1** (Properties of $\mathcal{P}_T$ and $\mathcal{P}_{T\perp}$ )**.**

$$\langle \mathcal{P}_T(X), Y \rangle = \langle X, \mathcal{P}_T(Y) \rangle = \langle \mathcal{P}_T(X), \mathcal{P}_T(Y) \rangle$$

$$\langle \mathcal{P}_{T\perp}(X), Y \rangle = \langle X, \mathcal{P}_{T\perp}(Y) \rangle = \langle \mathcal{P}_{T\perp}(X), \mathcal{P}_{T\perp}(Y) \rangle$$

*Proof.* Using the property of inner product $\langle X, Y \rangle = \langle X^T, Y^T \rangle$ and definition of adjoint operator $\langle AX, Y \rangle = \langle X, A^*Y \rangle$, we have

$$\begin{aligned}
\langle \mathcal{P}_T(X), Y \rangle &= \langle UU^T X, Y \rangle + \langle XVV^T, Y \rangle - \langle UU^T XVV^T, Y \rangle \\
&= \langle UU^T X, Y \rangle + \langle VV^T X^T, Y^T \rangle - \langle VV^T X^T, (UU^T Y)^T \rangle \\
&= \langle X, UU^T Y \rangle + \langle X^T, VV^T Y^T \rangle - \langle X^T, VV^T Y^T UU^T \rangle \\
&= \langle X, UU^T Y \rangle + \langle X, YVV^T \rangle - \langle X, UU^T YVV^T \rangle \\
&= \langle X, \mathcal{P}_T(Y) \rangle.
\end{aligned}$$

Use the equality with $X = X, Y = \mathcal{P}_T(Y)$, we get

$$\langle X, \mathcal{P}_T(\mathcal{P}_T(Y)) \rangle = \langle \mathcal{P}_T(X), \mathcal{P}_T(Y) \rangle.$$

The result for $\mathcal{P}_{T^\perp}$ is the same as the third term in the previous derivation as $I - UU^T$ and $I - VV^T$ are both projection matrices that are self-adjoint. $\square$

In addition, given index set $D$, we define projection $\mathcal{P}_D$, such that

$$\mathcal{P}_D(X) = \begin{cases} [\mathcal{P}_D(X)]_{ij} = X_{ij}, & \text{if } (i,j) \in D; \\ [\mathcal{P}_D(X)]_{ij} = 0, & \text{Otherwise.} \end{cases}$$

For example, when $D = \{(i,j)|i = j\}$, $\mathcal{P}_D(X) = 0 \Leftrightarrow \text{diag}(X) = 0$.

Consider general convex optimization problem

$$\begin{aligned}
&\min_{C_1, C_2} \ \|C_1\|_* + \lambda \|C_2\|_1 \\
&s.t. \quad B = AC_1, \quad C_1 = C_2, \quad \mathcal{P}_D(C_1) = 0
\end{aligned} \tag{A.1}$$

where $A \in R^{n \times m}$ is arbitrary dictionary and $B \in R^{n \times N}$ is data samples. Note that when $B = X$, $A = X$, (A.1) is exactly (1).

**Lemma A.1.2.** *For optimization problem* (A.1)*, if we have a quadruplet* $(C, \Lambda_1, \Lambda_2, \Lambda_3)$ *where* $C_1 = C_2 = C$ *is feasible, the support* $\text{supp}(C) = \Omega \subseteq \tilde{\Omega}$*,* $\text{rank}(C) = r$ *and skinny SVD of* $C = U\Sigma V^T$ *($\Sigma$ is an $r \times r$ diagonal matrix and $U$, $V$ are of compatible size), moreover if* $\Lambda_1$*,* $\Lambda_2$*,* $\Lambda_3$ *satisfy*

① $\mathcal{P}_T(A^T \Lambda_1 - \Lambda_2 - \Lambda_3) = UV^T$      ③ $[\Lambda_2]_\Omega = \lambda \text{sgn}([C]_\Omega)$      ⑤ $[\Lambda_2]_{\tilde{\Omega}^c} < \lambda$

② $\|\mathcal{P}_{T^\perp}(A^T \Lambda_1 - \Lambda_2 - \Lambda_3)\| \le 1$      ④ $[\Lambda_2]_{\Omega^c \cap \tilde{\Omega}} \le \lambda$      ⑥ $\mathcal{P}_{D^c}(\Lambda_3) = 0$

*then all optimal solutions to* (A.1) *satisfy* $\text{supp}(C) \subseteq \tilde{\Omega}$*.*

*Proof.* The subgradient of $\|C\|_*$ is $UV^T + W_1$ for any $W_1 \in T^\perp$ and $\|W_1\| \le 1$. For any optimal solution $C^*$ we may choose $W_1$ such that $\|W_1\| = 1$, $\langle W_1, \mathcal{P}_{T^\perp} C^* \rangle = \|\mathcal{P}_{T^\perp} C^*\|_*$. Then by the definition of subgradient, convex function $\|C\|_*$ obey

$$\begin{aligned}
\|C^*\|_* &\ge \|C\|_* + \langle UV^T + W_1, C^* - C \rangle \\
&= \langle UV^T, \mathcal{P}_T(C^* - C) \rangle + \langle UV^T, \mathcal{P}_{T^\perp}(C^* - C) \rangle + \langle W_1, C^* - C \rangle \\
&= \langle UV^T, \mathcal{P}_T(C^* - C) \rangle + \|\mathcal{P}_{T^\perp} C^*\|_*.
\end{aligned} \tag{A.2}$$

To see the equality, note that $\langle UV^T, \mathcal{P}_{T^\perp}(A) \rangle = 0$ for any compatible matrix $A$ and the following identity that follows directly from the construction of $W_1$ and Lemma A.1.1

$$\langle W_1, C^* - C \rangle = \langle \mathcal{P}_{T^\perp} W_1, C^* - C \rangle = \langle W_1, \mathcal{P}_{T^\perp}(C^* - C) \rangle = \langle W_1, \mathcal{P}_{T^\perp} C^* \rangle = \|\mathcal{P}_{T^\perp} C^*\|_*.$$

Similarly, the subgradient of $\lambda \|C\|_1$ is $\lambda \text{sgn}(C) + W_2$, for any $W_2$ obeying $\text{supp}(W_2) \subseteq \Omega^c$ and $\|W_2\|_\infty \le \lambda$. We may choose $W_2$ such that $\|W_2\|_\infty = \lambda$ and $\langle [W_2]_{\Omega^c}, C^*_{\Omega^c} \rangle = \|C^*_{\Omega^c}\|_1$, then by the convexity of one norm,

$$\lambda \|C^*\|_1 \ge \lambda \|C\|_1 + \lambda \langle \partial \|C\|_1, C^* - C \rangle = \lambda \|C\|_1 + \langle \lambda \text{sgn}(C_\Omega), C^*_\Omega - C_\Omega \rangle + \lambda \|C^*_{\Omega^c}\|_1. \tag{A.3}$$

Then we may combine (A.2) and (A.3) with condition ① and ③ to get

$$\|C^*\|_* + \lambda\|C^*\|_1$$
$$\geq \|C\|_* + \langle UV^T, \mathcal{P}_T(C^* - C)\rangle + \|\mathcal{P}_{T^\perp}(C^*)\|_* + \lambda\|C\|_1$$
$$+ \langle \lambda\mathrm{sgn}(C_\Omega), C_\Omega^* - C_\Omega\rangle + \lambda\|C_{\Omega^c}^*\|_1$$
$$= \|C\|_* + \langle \mathcal{P}_T(A^T\Lambda_1 - \Lambda_2 - \Lambda_3), \mathcal{P}_T(C^* - C)\rangle + \|\mathcal{P}_{T^\perp}(C^*)\|_* + \lambda\|C\|_1$$
$$+ \langle \Lambda_2, C_\Omega^* - C_\Omega\rangle + \lambda\|C_{\Omega^c \cap \tilde\Omega}^*\|_1 + \lambda\|C_{\tilde\Omega^c}^*\|_1. \tag{A.4}$$

By Lemma A.1.1, we know

$$\langle \mathcal{P}_T(A^T\Lambda_1 - \Lambda_2 - \Lambda_3), \mathcal{P}_T(C^* - C)\rangle$$
$$= \langle A^T\Lambda_1 - \Lambda_2 - \Lambda_3, \mathcal{P}_T(\mathcal{P}_T(C^* - C))\rangle$$
$$= \langle A^T\Lambda_1 - \Lambda_2 - \Lambda_3, \mathcal{P}_T(C^*)\rangle - \langle A^T\Lambda_1 - \Lambda_2 - \Lambda_3, \mathcal{P}_T(C)\rangle$$
$$= \langle \Lambda_1, A\mathcal{P}_T(C^*)\rangle - \langle \Lambda_2 + \Lambda_3, \mathcal{P}_T(C^*)\rangle - \langle \Lambda_1, AC\rangle + \langle \Lambda_2 + \Lambda_3, C\rangle$$
$$= \langle \Lambda_1, AC^* - AC\rangle - \langle \Lambda_1, A\mathcal{P}_{T^\perp}(C^*)\rangle + \langle \Lambda_2 + \Lambda_3, C\rangle - \langle \Lambda_2 + \Lambda_3, \mathcal{P}_T(C^*)\rangle$$
$$= -\langle \Lambda_1, A\mathcal{P}_{T^\perp}(C^*)\rangle + \langle \Lambda_2 + \Lambda_3, C\rangle - \langle \Lambda_2 + \Lambda_3, C^*\rangle + \langle \Lambda_2 + \Lambda_3, \mathcal{P}_{T^\perp}(C^*)\rangle$$
$$= -\langle A^T\Lambda_1 - \Lambda_2 - \Lambda_3, \mathcal{P}_{T^\perp}(C^*)\rangle - \langle \Lambda_2 + \Lambda_3, C^*\rangle + \langle \Lambda_2 + \Lambda_3, C\rangle$$
$$= -\langle \mathcal{P}_{T^\perp}(A^T\Lambda_1 - \Lambda_2), \mathcal{P}_{T^\perp}(C^*)\rangle - \langle \Lambda_2 + \Lambda_3, C^*\rangle + \langle \Lambda_2 + \Lambda_3, C\rangle$$
$$= -\langle \mathcal{P}_{T^\perp}(A^T\Lambda_1 - \Lambda_2), \mathcal{P}_{T^\perp}(C^*)\rangle - \langle \Lambda_2, C^*\rangle + \langle \Lambda_2, C\rangle.$$

Note that the last step follows from condition ⑥ and $C$, $C^*$'s primal feasibility. Substitute back into (A.4), we get

$$\|C^*\|_* + \lambda\|C^*\|_1$$
$$\geq \|C\|_* + \lambda\|C\|_1 + \|\mathcal{P}_{T^\perp}(C^*)\|_* - \langle \mathcal{P}_{T^\perp}(A^T\Lambda_1 - \Lambda_2 - \Lambda_3), \mathcal{P}_{T^\perp}(C^*)\rangle$$
$$+ \lambda\|C_{\Omega^c \cap \tilde\Omega}^*\|_1 - \langle [\Lambda_2]_{\Omega^c \cap \tilde\Omega}, C_{\Omega^c \cap \tilde\Omega}^*\rangle + \lambda\|C_{\tilde\Omega^c}^*\|_1 - \langle [\Lambda_2]_{\tilde\Omega^c}, C_{\tilde\Omega^c}^*\rangle$$
$$\geq \|C\|_* + \lambda\|C\|_1 - (1 - \|\mathcal{P}_{T^\perp}(A^T\Lambda_1 - \Lambda_2 - \Lambda_3)\|)\|\mathcal{P}_{T^\perp}(C^*)\|_*$$
$$(\lambda - \|[\Lambda_2]_{\Omega^c \cap \tilde\Omega}\|_\infty)\|C_{\Omega^c \cap \tilde\Omega}^*\|_1 + (\lambda - \|[\Lambda_2]_{\tilde\Omega^c}\|_\infty)\|C_{\tilde\Omega^c}^*\|_1$$

Assume $C_{\tilde\Omega^c}^* \neq 0$. By condition ④, ⑤ and ②, we have the strict inequality

$$\|C^*\|_* + \lambda\|C^*\|_1 > \|C\|_* + \lambda\|C\|_1.$$

Recall that $C^*$ is an optimal solution, i.e., $\|C^*\|_* + \lambda\|C^*\|_1 \leq \|C\|_* + \lambda\|C\|_1$. By contradiction, we conclude that $C_{\tilde\Omega^c}^* = 0$ for any optimal solution $C^*$. □

### A.2 Constructing solution

Apply Lemma A.1.2 with $A = X$, $B = X$ and $\tilde\Omega$ is selected such that the Self-Expressiveness Property (SEP) holds, then if we can find $\Lambda_1$ and $\Lambda_2$ satisfying the five conditions with respect to a feasible $C$, then we know all optimal solutions of (1) obey SEP. The dimension of the dual variables are $\Lambda_1 \in \mathbb{R}^{n \times N}$ and $\Lambda_2 \in \mathbb{R}^{N \times N}$.

**First layer fictitious problem**

A good candidate can be constructed by the optimal solutions of the fictitious programs for $i = 1, 2, 3$

$$\mathbf{P_1}: \quad \min_{C_1^{(i)}, C_2^{(i)}} \|C_1^{(i)}\|_* + \lambda\|C_2^{(i)}\|_1$$
$$\text{s.t.} \quad X^{(i)} = XC_1^{(i)}, \ C_1^{(i)} = C_2^{(i)}, \ \mathcal{P}_{D_i}(C_1^{(i)}) = 0. \tag{A.5}$$

Corresponding dual problem is

$$\mathbf{D_1}: \quad \max_{\Lambda_1^{(i)}, \Lambda_2^{(i)}, \Lambda_3^{(i)}} \langle X^{(i)}, \Lambda_1^{(i)}\rangle$$
$$\text{s.t.} \quad \|\Lambda_2^{(i)}\|_\infty \leq \lambda, \ \|X^T\Lambda_1^{(i)} - \Lambda_2^{(i)} - \Lambda_3^{(i)}\| \leq 1, \ \mathcal{P}_{D_i^c}(\Lambda_3^{(i)}) = 0 \tag{A.6}$$

where $\Lambda_1^{(i)} \in \mathbb{R}^{n \times N_i}$ and $\Lambda_2^{(i)}, \Lambda_3^{(i)} \in \mathbb{R}^{N \times N_i}$. $D_i$ is the diagonal set of the $i^{th}$ $N_i \times N_i$ block of $C_1^{(i)}$. For instance for $i = 2$,

$$C_1^{(2)} = \begin{pmatrix} 0 \\ \tilde{C}_1^{(2)} \\ 0 \end{pmatrix}, \qquad D_2 = \left\{ (i,j) \, \middle| \, \begin{bmatrix} 0 \\ I \\ 0 \end{bmatrix}_{ij} \neq 0 \right\},$$

The candidate solution is $C = \begin{bmatrix} C_1^{(1)} & C_1^{(2)} & C_1^{(3)} \end{bmatrix}$. Now we need to use a second layer of fictitious problem and the same Lemma A.1.2 with $A = X$, $B = X^{(i)}$ to show that the solution support $\tilde{\Omega}^{(i)}$ is like the following

$$C_1^{(1)} = \begin{pmatrix} \tilde{C}_1^{(1)} \\ 0 \\ 0 \end{pmatrix}, \quad C_1^{(2)} = \begin{pmatrix} 0 \\ \tilde{C}_1^{(2)} \\ 0 \end{pmatrix}, \quad C_1^{(3)} = \begin{pmatrix} 0 \\ 0 \\ \tilde{C}_1^{(3)} \end{pmatrix}. \tag{A.7}$$

**Second layer fictitious problem**

The second level of fictitious problems are used to construct a suitable solution. Consider for $i = 1, 2, 3$,

$$\mathbf{P_2}: \quad \min_{\tilde{C}_1^{(i)}, \tilde{C}_2^{(i)}} \|\tilde{C}_1^{(i)}\|_* + \lambda \|\tilde{C}_2^{(i)}\|_1 \tag{A.8}$$

$$s.t. \quad X^{(i)} = X^{(i)} \tilde{C}_1^{(i)}, \; \tilde{C}_1^{(i)} = \tilde{C}_2^{(i)}, \; \mathrm{diag}(\tilde{C}_1^{(i)}) = 0.$$

which is apparently feasible. Note that the only difference between the second layer fictitious problem (A.8) and the first layer fictitious problem (A.5) is the dictionary/design matrix being used. In (A.5), the dictionary contains all data points, whereas here in (A.8), the dictionary is nothing but $X^{(i)}$ itself. The corresponding dimension of representation matrix $C_1^{(i)}$ and $\tilde{C}_1^{(i)}$ are of course different too. Sufficiently we hope to establish the conditions where the solutions of (A.8) and (A.5) are related by (A.7).

The corresponding dual problem is

$$\mathbf{D_2}: \quad \max_{\tilde{\Lambda}_1^{(i)}, \tilde{\Lambda}_2^{(i)}, \tilde{\Lambda}_3^{(i)}} \langle X^{(i)}, \tilde{\Lambda}_1^{(i)} \rangle \tag{A.9}$$

$$s.t. \quad \|\tilde{\Lambda}_2^{(i)}\|_\infty \leq \lambda, \; \|[X^{(i)}]^T \tilde{\Lambda}_1^{(i)} - \tilde{\Lambda}_2^{(i)} - \tilde{\Lambda}_3^{(i)}\| \leq 1, \; \mathrm{diag}^\perp \left( \tilde{\Lambda}_3^{(i)} \right) = 0$$

where $\tilde{\Lambda}_1^{(i)} \in \mathbb{R}^{n \times N_i}$ and $\tilde{\Lambda}_2^{(i)}, \tilde{\Lambda}_3^{(i)} \in \mathbb{R}^{N_i \times N_i}$.

The proof is two steps. First we show the solution of (A.8), zero padded as in (A.7) are indeed optimal solutions of (A.5) and verify that all optimal solutions have such shape using Lemma A.1.2. The second step is to verify that solution $C = \begin{bmatrix} C_1^{(1)} & C_1^{(2)} & C_1^{(3)} \end{bmatrix}$ is optimal solution of (1).

### A.3 Constructing dual certificates

To complete the first step, we need to construct $\Lambda_1^{(i)}$, $\Lambda_2^{(i)}$ and $\Lambda_3^{(i)}$ such that all conditions in Lemma A.1.2 are satisfied. We use $i = 1$ to illustrate. Let the optimal solution[1] of (A.9) be $\tilde{\Lambda}_1^{(1)}$, $\tilde{\Lambda}_2^{(1)}$ and $\tilde{\Lambda}_3^{(1)}$. We set $\Lambda_1^{(1)} = \tilde{\Lambda}_1^{(1)}$, $\Lambda_2^{(1)} = \begin{pmatrix} \tilde{\Lambda}_2^{(1)} \\ \Lambda_a \\ \Lambda_b \end{pmatrix}$ and $\Lambda_3^{(1)} = \begin{pmatrix} \tilde{\Lambda}_3^{(1)} \\ 0 \\ 0 \end{pmatrix}$. As $\tilde{\Omega}$ defines the first block now, this construction naturally guarantees ③ and ④. ⑥ follows directly from the dual feasibility. The existence of $\Lambda_a$ and $\Lambda_b$ obeying ⑤①② is something we need to show.

To evaluate ① and ②, let's first define the projection operator. Take skinny SVD $\tilde{C}_1^{(1)} = \tilde{U}^{(1)} \tilde{\Sigma}^{(1)} (\tilde{V}^{(1)})^T$.

$$C_1^{(1)} = \begin{pmatrix} \tilde{C}_1^{(1)} \\ 0 \\ 0 \end{pmatrix} = \begin{pmatrix} \tilde{U}^{(1)} \\ 0 \\ 0 \end{pmatrix} \tilde{\Sigma}^{(1)} (\tilde{V}^{(1)})^T$$

$$U^{(1)}[U^{(1)}]^T = \begin{pmatrix} \tilde{U}^{(1)}[\tilde{U}^{(1)}]^T & 0 & 0 \\ 0 & 0 & 0 \\ 0 & 0 & 0 \end{pmatrix}, \qquad V^{(1)}[V^{(1)}]^T = \tilde{V}^{(1)}(\tilde{V}^{(1)})^T$$

For condition ① we need

$$\mathcal{P}_{T_1}\left( X^T \Lambda_1^{(1)} - \Lambda_2^{(1)} \right) = \mathcal{P}_{T_1} \begin{pmatrix} [X^{(1)}]^T \tilde{\Lambda}_1^{(1)} - \tilde{\Lambda}_2 - \tilde{\Lambda}_3 \\ [X^{(2)}]^T \tilde{\Lambda}_1^{(1)} - \Lambda_a \\ [X^{(3)}]^T \tilde{\Lambda}_1^{(1)} - \Lambda_b \end{pmatrix}$$

$$= \begin{pmatrix} \mathcal{P}_{\tilde{T}_1}([X^{(1)}]^T \tilde{\Lambda}_1^{(1)} - \tilde{\Lambda}_2 - \tilde{\Lambda}_3) \\ ([X^{(2)}]^T \tilde{\Lambda}_1^{(1)} - \Lambda_a)\tilde{V}^{(1)}(\tilde{V}^{(1)})^T \\ ([X^{(3)}]^T \tilde{\Lambda}_1^{(1)} - \Lambda_b)\tilde{V}^{(1)}(\tilde{V}^{(1)})^T \end{pmatrix} = \begin{pmatrix} \tilde{U}^{(1)}[\tilde{V}^{(1)}]^T \\ 0 \\ 0 \end{pmatrix}$$

The first row is guaranteed by construction. The second and third row are something we need to show. For condition ②

$$\left\| \mathcal{P}_{T_1^\perp}\left( X^T \Lambda_1^{(1)} - \Lambda_2^{(1)} - \tilde{\Lambda}_3 \right) \right\| = \left\| \begin{pmatrix} \mathcal{P}_{\tilde{T}_1^\perp}([X^{(1)}]^T \tilde{\Lambda}_1^{(1)} - \tilde{\Lambda}_2 - \tilde{\Lambda}_3) \\ ([X^{(2)}]^T \tilde{\Lambda}_1^{(1)} - \Lambda_a)(I - \tilde{V}^{(1)}(\tilde{V}^{(1)})^T) \\ ([X^{(3)}]^T \tilde{\Lambda}_1^{(1)} - \Lambda_b)(I - \tilde{V}^{(1)}(\tilde{V}^{(1)})^T) \end{pmatrix} \right\|$$

$$\leq \|\mathcal{P}_{\tilde{T}_1^\perp}([X^{(1)}]^T \tilde{\Lambda}_1^{(1)} - \tilde{\Lambda}_2 - \tilde{\Lambda}_3)\| + \|[X^{(2)}]^T \tilde{\Lambda}_1^{(1)} - \Lambda_a\| + \|[X^{(3)}]^T \tilde{\Lambda}_1^{(1)} - \Lambda_b\|$$

Note that as $([X^{(2)}]^T \tilde{\Lambda}_1^{(1)} - \Lambda_a)\tilde{V}^{(1)}(\tilde{V}^{(1)})^T = 0$, the complement projection $([X^{(2)}]^T \tilde{\Lambda}_1^{(1)} - \Lambda_a)(I - \tilde{V}^{(1)}(\tilde{V}^{(1)})^T) = ([X^{(2)}]^T \tilde{\Lambda}_1^{(1)} - \Lambda_a)$. The same goes for the third row. In fact, in worst case, $\|\mathcal{P}_{\tilde{T}_1^\perp}([X^{(1)}]^T \tilde{\Lambda}_1^{(1)} - \tilde{\Lambda}_2)\| = 1$, then for both ① and ② to hold, we need

$$[X^{(2)}]^T \tilde{\Lambda}_1^{(1)} - \Lambda_a = 0, \qquad\qquad [X^{(3)}]^T \tilde{\Lambda}_1^{(1)} - \Lambda_b = 0. \qquad (A.10)$$

In other words, the conditions reduce to whether there exist $\Lambda_a, \Lambda_b$ obeying $\|\Lambda_a\|_\infty < \lambda$ and $\|\Lambda_b\|_\infty < \lambda$ that can nullify $[X^{(2)}]^T \tilde{\Lambda}_1^{(1)}$ and $[X^{(3)}]^T \tilde{\Lambda}_1^{(1)}$.

In fact, as we will illustrate, (A.10) is sufficient for the original optimization (1) too. We start the argument by taking the skinny SVD of constructed solution $C$.

$$C = \begin{pmatrix} \tilde{C}_1 & 0 & 0 \\ 0 & \tilde{C}_2 & 0 \\ 0 & 0 & \tilde{C}_3 \end{pmatrix} = \begin{pmatrix} \tilde{U}_1 & 0 & 0 \\ 0 & \tilde{U}_2 & 0 \\ 0 & 0 & \tilde{U}_3 \end{pmatrix} \begin{pmatrix} \tilde{\Sigma}_1 & 0 & 0 \\ 0 & \tilde{\Sigma}_2 & 0 \\ 0 & 0 & \tilde{\Sigma}_3 \end{pmatrix} \begin{pmatrix} \tilde{V}_1 & 0 & 0 \\ 0 & \tilde{V}_2 & 0 \\ 0 & 0 & \tilde{V}_3 \end{pmatrix}.$$

Check that $U, V$ are both orthonormal, $\Sigma$ is diagonal matrix with unordered singular values. Let the block diagonal shape be $\Omega$, the five conditions in Lemma A.1.2 are met with

$$\Lambda_1 = \begin{pmatrix} \tilde{\Lambda}_1^{(1)} & \tilde{\Lambda}_1^{(2)} & \tilde{\Lambda}_1^{(3)} \end{pmatrix}, \quad \Lambda_2 = \begin{pmatrix} \tilde{\Lambda}_2^{(1)} & \Lambda_a^{(2)} & \Lambda_a^{(3)} \\ \Lambda_a^{(1)} & \tilde{\Lambda}_2^{(2)} & \Lambda_b^{(3)} \\ \Lambda_b^{(1)} & \Lambda_b^{(2)} & \tilde{\Lambda}_2^{(3)} \end{pmatrix}, \quad \Lambda_3 = \begin{pmatrix} \tilde{\Lambda}_3^{(1)} & 0 & 0 \\ 0 & \tilde{\Lambda}_3^{(2)} & 0 \\ 0 & 0 & \tilde{\Lambda}_3^{(3)} \end{pmatrix},$$

as long as $\Lambda_1^{(i)}, \Lambda_2^{(i)}$ and $\Lambda_3^{(i)}$ guarantee the optimal solution of (A.5) obeys SEP for each $i$. Condition ③ ④ ⑤ and ⑥ are trivial. To verify condition ① and ②,

$$X^T \Lambda_1 - \Lambda_2 - \Lambda_3$$

$$= \begin{pmatrix} [X^{(1)}]^T \tilde{\Lambda}_1^{(1)} - \tilde{\Lambda}_2^{(1)} - \tilde{\Lambda}_3^{(1)} & [X^{(1)}]^T \tilde{\Lambda}_1^{(2)} - \Lambda_a^{(2)} & [X^{(1)}]^T \tilde{\Lambda}_1^{(3)} - \Lambda_a^{(3)} \\ [X^{(2)}]^T \tilde{\Lambda}_1^{(1)} - \Lambda_a^{(1)} & [X^{(2)}]^T \tilde{\Lambda}_1^{(2)} - \tilde{\Lambda}_2^{(2)} - \tilde{\Lambda}_3^{(2)} & [X^{(2)}]^T \tilde{\Lambda}_1^{(3)} - \Lambda_b^{(3)} \\ [X^{(3)}]^T \tilde{\Lambda}_1^{(1)} - \Lambda_b^{(1)} & [X^{(3)}]^T \tilde{\Lambda}_1^{(2)} - \Lambda_b^{(2)} & [X^{(3)}]^T \tilde{\Lambda}_1^{(3)} - \tilde{\Lambda}_2^{(3)} - \tilde{\Lambda}_3^{(3)} \end{pmatrix}$$

$$= \begin{pmatrix} [X^{(1)}]^T \tilde{\Lambda}_1^{(1)} - \tilde{\Lambda}_2^{(1)} - \tilde{\Lambda}_3^{(1)} & 0 & 0 \\ 0 & [X^{(2)}]^T \tilde{\Lambda}_1^{(2)} - \tilde{\Lambda}_2^{(2)} - \tilde{\Lambda}_3^{(2)} & 0 \\ 0 & 0 & [X^{(3)}]^T \tilde{\Lambda}_1^{(3)} - \tilde{\Lambda}_2^{(3)} - \tilde{\Lambda}_3^{(3)} \end{pmatrix}.$$

Furthermore, by the block-diagonal SVD of $C$, projection $\mathcal{P}_T$ can be evaluated for each diagonal block, where optimality condition of the second layer fictitious problem guarantees that for each $i$

$$\mathcal{P}_{\tilde{T}_i}([X^{(i)}]^T \tilde{\Lambda}_1^{(i)} - \tilde{\Lambda}_2^{(i)} - \tilde{\Lambda}_3^{(i)}) = \tilde{U}_i \tilde{V}_i^T.$$

It therefore holds that

① $\quad \mathcal{P}_T(X^T\Lambda_1 - \Lambda_2 - \Lambda_3) = \begin{pmatrix} \tilde{U}_1\tilde{V}_1^T & 0 & 0 \\ 0 & \tilde{U}_2\tilde{V}_2^T & 0 \\ 0 & 0 & \tilde{U}_3\tilde{V}_3^T \end{pmatrix} = UV^T,$

② $\quad \|\mathcal{P}_{T^\perp}(X^T\Lambda_1 - \Lambda_2)\|$

$$= \left\| \begin{array}{ccc} \mathcal{P}_{\tilde{T}_i^\perp}([X^{(1)}]^T \tilde{\Lambda}_1^{(1)} - \tilde{\Lambda}_2^{(1)}) & 0 & 0 \\ 0 & \mathcal{P}_{\tilde{T}_i^\perp}([X^{(2)}]^T \tilde{\Lambda}_1^{(2)} - \tilde{\Lambda}_2^{(2)}) & 0 \\ 0 & 0 & \mathcal{P}_{\tilde{T}_i^\perp}([X^{(3)}]^T \tilde{\Lambda}_1^{(3)} - \tilde{\Lambda}_2^{(3)}) \end{array} \right\|$$

$$= \max_{i=1,2,3} \|\mathcal{P}_{\tilde{T}_i^\perp}([X^{(1)}]^T \tilde{\Lambda}_1^{(i)} - \tilde{\Lambda}_2^{(i)})\| \leq 1.$$

## A.4 Dual Separation Condition

**Definition A.4.1** (Dual Separation Condition). *For $X^{(i)}$, if the corresponding dual optimal solution $\tilde{\Lambda}_1^{(i)}$ of (A.9) obeys $\|[X^{(j)}]^T \tilde{\Lambda}_1^{(i)}\|_\infty < \lambda$ for all $j \neq i$, then we say that dual separation condition holds.*

**Remark A.4.1.** Definition A.4.1 directly implies the existence of $\Lambda_a$, $\Lambda_b$ obeying (A.10).

Bounding $\|[X^{(j)}]^T \tilde{\Lambda}_1^{(i)}\|_\infty$ is equivalent to bound the maximal inner product of arbitrary column pair of $X^{(j)}$ and $\tilde{\Lambda}_1^{(i)}$. Let $x$ be a column of $X^{(j)}$ and $\nu$ be a column of $\tilde{\Lambda}_1^{(i)}$,

$$\langle x, \nu \rangle = \|\nu^*\| \langle x, \frac{\nu}{\|\nu^*\|} \rangle \leq \|\nu^*\| \|[V^{(i)}]^T x\|_\infty \leq \max_k \|\mathrm{Proj}_{\mathcal{S}_i}(\tilde{\Lambda}_1^{(i)}) \mathbf{e}_k\| \max_{x \in \mathcal{X} \backslash \mathcal{X}_i} \|[V^{(i)}]^T x\|_\infty.$$

where $V^{(i)} = [\frac{\nu_1}{\|\nu_1^*\|}, ..., \frac{\nu_{N_i}}{\|\nu_{N_i}^*\|}]$ is a normalized dual matrix as defined in Definition 2 and $\mathbf{e}_k$ denotes standard basis. Recall that in Definition 2, $\nu^*$ is the component of $\nu$ inside $\mathcal{S}_i$ and $\nu$ is normalized such that $\|\nu^*\| = 1$. It is easy to verify that $[\tilde{\Lambda}_1^{(i)}]^* = \mathrm{Proj}_{\mathcal{S}_i}(\tilde{\Lambda}_1^{(i)})$ is minimum-Frobenious-norm optimal solution. Note that we can choose $\tilde{\Lambda}_1^{(i)}$ to be any optimal solution of (A.9), so we take $\tilde{\Lambda}_1^{(i)}$ such that the associated $V^{(i)}$ is the one that minimizes $\max_{x \in \mathcal{X} \backslash \mathcal{X}_i} \|[V^{(i)}]^T x\|_\infty$.

Now we may write a sufficient dual separation condition in terms of the incoherence $\mu$ in Definition 3,

$$\langle x, \nu \rangle \leq \max_k \|[\tilde{\Lambda}_1^{(i)}]^* \mathbf{e}_k\| \mu(\mathcal{X}_i) \leq \lambda. \tag{A.11}$$

Now it is left to bound $\max_k \|[\tilde{\Lambda}_1^{(i)}]^* \mathbf{e}_k\|$ with meaningful properties of $X^{(i)}$.

### A.4.1 Separation condition via singular value

By the second constraint of (A.9), we have

$$1 \geq \|[X^{(i)}]^T \tilde{\Lambda}_1^{(i)} - \tilde{\Lambda}_2^{(i)} - \tilde{\Lambda}_3^{(i)}\| \geq \max_k \|([X^{(i)}]^T \tilde{\Lambda}_1^{(i)} - \tilde{\Lambda}_2^{(i)} - \tilde{\Lambda}_3^{(i)}) \mathbf{e}_k\| := \|v\| \tag{A.12}$$

Note that $\max_k \|([X^{(i)}]^T \tilde{\Lambda}_1^{(i)} - \tilde{\Lambda}_2^{(i)} - \tilde{\Lambda}_3^{(i)}) \mathbf{e}_k\|$ is the 2-norm of a vector and we conveniently denote this vector by $v$. It follows that

$$\|v\| = \sqrt{|v_k|^2 + \sum_{i \neq k} |v_i|^2} \geq \sqrt{\sum_{i \neq k} |v_i|^2} = \|v_{-k}\|, \tag{A.13}$$

where $v_k$ denotes the $k^{th}$ element and $v_{-k}$ stands for $v$ with the $k^{th}$ element removed. For convenience, we also define $X_{-k}$ to be $X$ with the $k^{th}$ column removed and $X_k$ to be the $k^{th}$ column vector of $X$.

By condition ⑥ in Lemma A.1.2, $\tilde{\Lambda}_3^{(i)}$ is diagonal, hence $\tilde{\Lambda}_3^{(i)}\mathbf{e}_k = \left[0,...,[\tilde{\Lambda}_3^{(i)}\mathbf{e}_k]_k,...,0\right]^T$ and $[\tilde{\Lambda}_3^{(i)}\mathbf{e}_k]_{-k} = 0$. To be precise, we may get rid of $\tilde{\Lambda}_3^{(i)}$ all together

$$\|v_{-k}\| = \max_k \left\| \left([X_{-k}^{(i)}]^T \tilde{\Lambda}_1^{(i)} - [[\tilde{\Lambda}_2^{(i)}]^T]_{-k}\right)\mathbf{e}_k \right\|.$$

Note that $\max_k \|X\mathbf{e}_k\|$ is a norm, as is easily shown in the following lemma.

**Lemma A.4.1.** *Function $f(X) := \max_k \|X\mathbf{e}_k\|$ is a norm.*

*Proof.* We prove by definition of a norm.
(1) $f(aX) = \max_k \|[aX]_k\| = \max_k(|a|\|X_k\|) = \|a\| f(X)$.
(2) Assume $X \neq 0$ and $f(X) = 0$. Then for some $(i, j)$, $X_{ij} = c \neq 0$, so $f(X) \geq |c|$ which contradicts $f(X) = 0$.
(3) Triangular inequality:

$$f(X_1 + X_2) = \max_k(\|[X_1 + X_2]_k\|) \leq \max_k(\|[X_1]_k\| + \|[X_2]_k\|)$$
$$\leq \max_{k_1}(\|[X_1]_{k_1}\|) + \max_{k_2}(\|[X_2]_{k_2}\|) = f(X_1) + f(X_2).$$

$\square$

Thus by triangular inequality,

$$\|v_{-k}\| \geq \max_k \left\| [X_{-k}^{(i)}]^T[\tilde{\Lambda}_1^{(i)}\mathbf{e}_k] \right\| - \max_k \left\| [[\tilde{\Lambda}_2^{(i)}]^T]_{-k}\mathbf{e}_k \right\|$$
$$\geq \sigma_{d_i}(X_{-k}^{(i)})\max_k \|[\tilde{\Lambda}_1^{(i)}]^*\mathbf{e}_k\| - \lambda\sqrt{N_i - 1} \tag{A.14}$$

where $\sigma_{d_i}(X_{-k}^{(i)})$ is the $r^{th}$ (smallest non-zero) singular value of $X_{-k}^{(i)}$. The last inequality is true because $X_{-k}^{(i)}$ and $[\tilde{\Lambda}_1^{(i)}]^*$ belong to the same $d_i$-dimensional subspace and the condition $\|\tilde{\Lambda}_2^{(i)}\|_\infty \leq \lambda$. Combining (A.12)(A.13) and (A.14), we find the desired bound

$$\max_k \|[\tilde{\Lambda}_1^{(i)}]^*\mathbf{e}_k\| \leq \frac{1 + \lambda\sqrt{N_i - 1}}{\sigma_{d_i}(X_{-k}^{(i)})} < \frac{1 + \lambda\sqrt{N_i}}{\sigma_{d_i}(X_{-k}^{(i)})}.$$

The condition (A.11) now becomes

$$\langle x, \nu \rangle \leq \frac{\mu(1 + \lambda\sqrt{N_i})}{\sigma_{d_i}(X_{-k}^{(i)})} < \lambda \iff \mu(1 + \lambda\sqrt{N_i}) < \lambda\sigma_{d_i}(X_{-k}^{(i)}). \tag{A.15}$$

Note that when $X^{(i)}$ is well conditioned with condition number $\kappa$,

$$\sigma_{d_i}(X_{-k}^{(i)}) = \frac{1}{\kappa\sqrt{d_i}}\|X_{-k}^{(i)}\|_F = (1/\kappa)\sqrt{N_i/d_i}.$$

To interpret the inequality, we remark that when $\mu\kappa\sqrt{d_i} < 1$ there always exists a $\lambda$ such that SEP holds.

### A.4.2 Separation condition via inradius

This time we relax the inequality in (A.14) towards the max/infinity norm.

$$\|v_{-k}\| = \max_k \left\| \left([X_{-k}^{(i)}]^T\tilde{\Lambda}_1^{(i)} - [[\tilde{\Lambda}_2^{(i)}]^T]_{-k}\right)\mathbf{e}_k \right\|$$
$$\geq \max_k \left\| \left([X_{-k}^{(i)}]^T\tilde{\Lambda}_1^{(i)} - [[\tilde{\Lambda}_2^{(i)}]^T]_{-k}\right)\mathbf{e}_k \right\|_\infty$$
$$\geq \max_k \left\| [X_{-k}^{(i)}]^T[\tilde{\Lambda}_1^{(i)}]^* \right\|_\infty - \lambda \tag{A.16}$$

This is equivalent to for all $k = 1, .., N_i$

$$
\begin{cases}
\|[X_{-k}^{(i)}]^T \nu_1^*\|_\infty \leq 1 + \lambda, \\
\|[X_{-k}^{(i)}]^T \nu_2^*\|_\infty \leq 1 + \lambda, \\
... \\
\|[X_{-k}^{(i)}]^T \nu_{N_i}^*\|_\infty \leq 1 + \lambda,
\end{cases}
\Leftrightarrow
\begin{cases}
\nu_1^* \in (1 + \lambda)[\text{conv}(\pm X_{-k}^{(i)})]^o, \\
\nu_2^* \in (1 + \lambda)[\text{conv}(\pm X_{-k}^{(i)})]^o, \\
... \\
\nu_{N_i}^* \in (1 + \lambda)[\text{conv}(\pm X_{-k}^{(i)})]^o,
\end{cases}
$$

where $\mathcal{P}^o$ represents the polar set of a convex set $\mathcal{P}$, namely, every column of $\tilde{\Lambda}_1^{(i)}$ in (A.11) is within this convex polytope $[\text{conv}(\pm X_{-k}^{(i)})]^o$ scaled by $(1 + \lambda)$. A upper bound follows from the geometric properties of the symmetric convex polytope.

**Definition A.4.2** (circumradius). *The circumradius of a convex body $\mathcal{P}$, denoted by $R(\mathcal{P})$, is defined as the radius of the smallest Euclidean ball containing $\mathcal{P}$.*

The magnitude $\|\nu^*\|$ is bounded by $R([\text{conv}(\pm X_{-k}^{(i)})]^o)$. Moreover, by the the following lemma we may find the circumradius by analyzing the polar set of $[\text{conv}(\pm X_{-k}^{(i)})]^o$ instead. By the property of polar operator, polar of a polar set gives the tightest convex envelope of original set, i.e., $(\mathcal{K}^o)^o = \text{conv}(\mathcal{K})$. Since $\text{conv}(\pm X_{-k}^{(i)})$ is convex in the first place, the polar set is essentially $\text{conv}(\pm X_{-k}^{(i)})$.

**Lemma A.4.2.** *For a symmetric convex body $\mathcal{P}$, i.e. $\mathcal{P} = -\mathcal{P}$, inradius of $\mathcal{P}$ and circumradius of polar set of $\mathcal{P}$ satisfy:*

$$
r(\mathcal{P})R(\mathcal{P}^o) = 1.
$$

By this observation, we have for all $j = 1, ..., N_i$

$$
\|\nu_j^*\| \leq (1 + \lambda)R(\text{conv}(\pm X_{-k}^{(i)})) = \frac{1 + \lambda}{r(\text{conv}(\pm X_{-k}^{(i)}))}.
$$

Then the condition becomes

$$
\frac{\mu(1 + \lambda)}{r(\text{conv}(\pm X_{-k}^{(i)}))} < \lambda \; \Leftrightarrow \; \mu(1 + \lambda) < \lambda r(\text{conv}(\pm X_{-k}^{(i)})), \tag{A.17}
$$

which reduces to the condition of SSC when $\lambda$ is large (if we take the $\mu$ definition in [14]).

With (A.15) and (A.17), the proof for Theorem 1 is complete.

# B    Proof of Theorem 2 (the randomized result)

Theorem 2 is essentially a corollary of the deterministic results. The proof of it is no more than providing probabilistic lower bounds of smallest singular value $\sigma$ (Lemma 1), inradius (Lemma 2) and upper bounds for minimax subspace incoherence $\mu$ (Lemma 3), then use union bound to make sure all random events happen together with high probability.

## B.1    Smallest singular value of unit column random low-rank matrices

We prove Lemma 1 in this section. Assume the following mechanism of random matrix generation.

1. Generate $n \times r$ Gaussian random matrix $A$.
2. Generate $r \times N$ Gaussian random matrix $B$.
3. Generate rank-$r$ matrix $AB$ then normalize each column to unit vector to get $X$.

The proof contains three steps. First is to bound the magnitude. When $n$ is large, each column's magnitude is bounded from below with large probability. Second we show that if we reduce the largest magnitude column to smallest column vector, the singular values are only scaled by the same factor. Thirdly use singular value bound of $A$ and $B$ to show that singular value of $X$.

$$
2\sigma_r(X) > \sigma_r(AB) > \sigma_r(A)\sigma_r(B)
$$

**Lemma B.1.1** (Magnitude of Gaussian vector). *For Gaussian random vector $z \in \mathbb{R}^n$, if each entry $z_i \sim N(0, \frac{\sigma}{\sqrt{n}})$, then each column $z_i$ satisfies:*

$$Pr((1-t)\sigma^2 \leq \|z\|^2 \leq (1+t)\sigma^2) > 1 - e^{\frac{n}{2}(\log(t+1)-t)} - e^{\frac{n}{2}(\log(1-t)+t)}$$

*Proof.* To show the property, we observe that the sum of $n$ independent square Gaussian random variables follows $\chi^2$ distribution with d.o.f $n$, in other word, we have

$$\|z\|^2 = |z_1|^2 + ... + |z_n|^2 \sim \frac{\sigma^2}{n}\chi^2(n).$$

By Hoeffding's inequality, we have a close upper bound of its CDF [7], which gives us

$$Pr(\|z\|^2 > \alpha\sigma^2) = 1 - \text{CDF}_{\chi_n^2}(\alpha) \leq (\alpha e^{1-\alpha})^{\frac{n}{2}} \qquad \text{for } \alpha > 1,$$
$$Pr(\|z\|^2 < \beta\sigma^2) = \text{CDF}_{\chi_n^2}(\beta) \leq (\beta e^{1-\beta})^{\frac{n}{2}} \qquad \text{for } \beta < 1.$$

Substitute $\alpha = 1 + t$ and $\beta = 1 - t$, and apply union bound we get exactly the concentration statement. $\square$

To get an idea of the scale, when $t = 1/3$, the ratio of maximum and minimum $\|z\|$ is smaller than 2 with probability larger than $1 - 2\exp(-n/20)$. This proves the first step.

By random matrix theory [e.g., 12, 13, 8] asserts that $G$ is close to an orthonormal matrix, as the following lemma, adapted from Theorem II.13 of [8], shows:

**Lemma B.1.2** (Smallest singular value of random rectangular matrix). *Let $G \in \mathbb{R}^{n \times r}$ has i.i.d. entries $\sim N(0, 1/\sqrt{n})$. With probability of at least $1 - 2\gamma$,*

$$1 - \sqrt{\frac{r}{n}} - \sqrt{\frac{2\log(1/\gamma)}{n}} \leq \sigma_{min}(G) \leq \sigma_{max}(G) \leq 1 + \sqrt{\frac{r}{n}} + \sqrt{\frac{2\log(1/\gamma)}{n}}.$$

**Lemma B.1.3** (Smallest singular value of random low-rank matrix). *Let $A \in R^{n \times r}$, $B \in R^{r \times N}$, $r < N < n$, furthermore, $A_{ij} \sim N(0, 1/\sqrt{n})$ and $B_{ij} \sim N(0, 1/\sqrt{N})$. Then there exists an absolute constant $C$ such that with probability of at least $1 - n^{-10}$,*

$$\sigma_r(AB) \geq 1 - 3\sqrt{\frac{r}{N}} - C\sqrt{\frac{\log N_\ell}{N}}.$$

The proof is by simply by $\sigma_r(AB) \geq \sigma_r(A)\sigma_r(B)$, apply Lemma B.1.1 to both terms and then take $\gamma = \frac{1}{2N_\ell^{10}}$.

Now we may rescale each column of $AB$ to the maximum magnitude and get $\overline{AB}$. Naturally,

$$\sigma_r(\overline{AB}) \geq \sigma_r(AB).$$

On the other hand, by the results of Step 1,

$$\sigma_r(X) \geq \sigma_r(\underline{AB}) \geq \frac{1}{2}\sigma_r(\overline{AB}) \geq \frac{1}{2}\sigma_r(AB).$$

Normalizing the scale of the random matrix and plug in the above arguments, we get Lemma 1 in the main paper.

## B.2 Smallest inradius of random polytopes

This bound in Lemma 2 is due to Alonso-Gutiérrez in his proof of lower bound of the volume of a random polytope[1, Lemma 3.1]. The results was made clear in the subspace clustering context by Soltanokotabi and Candes[14, Lemma 7.4]. We refer the readers to the references for the proof.

### B.3 Upper bound of Minimax Subspace Incoherence

The upper bound of the minimax subspace incoherence (Lemma 3) we used in this paper is the same as the upper bound of the subspace incoherence in [14]. This is because for by taking $V = V^*$, the value will be larger by the minimax definition[2]. For completeness, we include the steps of proof here.

The argument critically relies on the following lemma on the area of spherical cap in [2].

**Lemma B.3.1** (Upper bound on the area of spherical cap). *Let $a \in \mathbb{R}^n$ be a random vector sampled from a unit sphere and $z$ is a fixed vector. Then we have:*

$$Pr\left(|a^T z| > \epsilon \|z\|\right) \leq 2e^{\frac{-n\epsilon^2}{2}}$$

With this result, Lemma 3 is proven in two steps. The first step is to apply Lemma B.3.1 to bound $\langle \nu_i^*, x \rangle$ and every data point $x \notin X^{(\ell)}$, where $\nu_i^*$ (a fixed vector) is the central dual vector corresponding to the data point $x_i \in X^{(\ell)}$ (see the Definition 3). When $\epsilon = \sqrt{\frac{6 \log(N)}{n}}$, the failure probability for one even is $\frac{2}{N^3}$. Recall that $\nu_i^*$ . The second step is to use union bound across all $x$ and then all $\nu_i^*$. The total number of events is less than $N^2$ so we get

$$\mu < \sqrt{\frac{6 \log N}{n}} \quad \text{with probability larger than } 1 - \frac{2}{N}.$$

### B.4 Bound of minimax subspace incoherence for semi-random model

Another bound of the subspace incoherence can be stated under the semi-random model in [14], where subspaces are deterministic and data in each subspaces are randomly sampled. The upper bound is given as a *log* term times the average cosine of the canonical angles between a pair of subspaces. This is not used in this paper, but the case of overlapping subspaces can be intuitively seen from the bound. The full statement is rather complex and is the same form as equation (7.6) of [14], so we refer the readers there for the full proof there and only include what is different from there: the proof that central dual vector $\nu_i^*$ distributes uniformly on the unit sphere of $\mathcal{S}_\ell$.

Let $U$ be a set of orthonormal basis of $\mathcal{S}_\ell$. Define rotation $R_{\mathcal{S}_\ell} := URU^T$ with arbitrary $d \times d$ rotation matrix $R$. If $\Lambda^*$ be the central optimal solution of (A.9), denoted by $\text{OptVal}(X^{(\ell)})$, it is easy to see that

$$R_{\mathcal{S}_\ell} \Lambda^* = \text{OptVal}(R_{\mathcal{S}_\ell} X^{(\ell)}).$$

Since $X^{(\ell)}$ distribute uniformly, the probability density of getting any $X^{(\ell)}$ is identical. For each fixed instance of $X^{(\ell)}$, consider $R$ a random variable, then the probability density of each column of $\Lambda^*$ be transformed to any direction is the same. Integrating the density over all different $X^{(\ell)}$, we completed the proof for the claim that the overall probability density of $\nu_i^*$ (each column of $\Lambda^*$) pointing towards any directions in $\mathcal{S}_\ell$ is the same.

Referring to [14], the upper bound is just a concentration bound saying that the smallest inner product is close to the average cosines of the canonical angles between two subspaces, which follows from the uniform distribution of $\nu_i^*$ and uniform distribution of $x$ in other subspaces. Therefore, when the dimension of each subspace is large, the average can still be small even though a small portion of the two subspaces are overlapping (a few canonical angles being equal to 1).

## C  Numerical algorithm

Like described in the main text, we will derive Alternating Direction Method of Multipliers (ADMM)[3] algorithm to solve LRSSC and NoisyLRSSC. We start from noiseless version then look at the noisy version.

**Algorithm 1** ADMM-LRSSC (with optional Adaptive Penalty)

---

**Input:** Data points as columns in $X \in \mathbb{R}^{n \times N}$, tradeoff parameter $\lambda$, numerical parameters $\mu_1^{(0)}, \mu_2^{(0)}, \mu_3^{(0)}$ and (optional $\rho_0, \mu_{max}, \eta, \epsilon$).
Initialize $C_1 = 0$, $C_2 = 0$, $J = 0$, $\Lambda_1 = 0$, $\Lambda_2 = 0$ and $\Lambda_3 = 0$.
Pre-compute $X^T X$ and $H = \left[ \mu_1 X^T X + (\mu_2 + \mu_3) I \right]^{-1}$ for later use.
**while** not converged **do**
    1. Update $J$ by (C.2).
    2. Update $C_1, C_2$ by (C.3).
    3. Update $\Lambda_1, \Lambda_2, \Lambda_3$ by (C.4).
    4. (Optional) Update parameter $(\mu_1, \mu_2, \mu_3) = \rho(\mu_1, \mu_2, \mu_3)$ and the pre-computed $H = H/\rho$
    where

$$\rho = \begin{cases} \min\left(\mu_{max}/\mu_1, \rho_0\right), & \text{if } \mu_1^{\text{prev}} \max(\sqrt{\eta}\|C_1 - C_1^{\text{prev}}\|_F)/\|X\|_F \le \epsilon; \\ 1, & \text{otherwise.} \end{cases}$$

**end while**
**Output:** Affinity matrix $W = |C_1| + |C_1|^T$

---

## C.1 ADMM for LRSSC

First we need to reformulate the optimization with two auxiliary terms, $C = C_1 = C_2$ as in the proof to separate the two norms, and $J$ to ensure each step has closed-form solution.

$$\begin{aligned} \min_{C_1, C_2, J} \quad & \|C_1\|_* + \lambda\|C_2\|_1 \\ s.t. \quad & X = XJ, \quad J = C_2 - \operatorname{diag}(C_2), \quad J = C_1 \end{aligned} \tag{C.1}$$

The Augmented Lagrangian is:

$$\begin{aligned} \mathcal{L} = & \|C_1\|_* + \lambda\|C_2\|_1 + \frac{\mu_1}{2}\|X - XJ\|_F^2 + \frac{\mu_2}{2}\|J - C_2 + \operatorname{diag}(C_2)\|_F^2 + \frac{\mu_3}{2}\|J - C_1\|_F^2 \\ & + tr(\Lambda_1^T(X - XJ)) + tr(\Lambda_2^T(J - C_2 + \operatorname{diag}(C_2))) + tr(\Lambda_3^T(J - C_1)), \end{aligned}$$

where $\mu_1$, $\mu_2$ and $\mu_3$ are numerical parameters to be tuned. By assigning the partial gradient/subgradient of $J$, $C_2$ and $C_1$ iteratively and update dual variables $\Lambda_1, \Lambda_2, \Lambda_3$ in every iterations, we obtain the update steps of ADMM.

$$J = \left[\mu_1 X^T X + (\mu_2 + \mu_3)I\right]^{-1}\left[\mu_1 X^T X + \mu_2 C_2 + \mu_3 C_1 + X^T \Lambda_1 - \Lambda_2 - \Lambda_3\right] \tag{C.2}$$

Define soft-thresholding operator $\pi_\beta(X) = (|X| - \beta)_+ \operatorname{sgn}(X)$ and singular value soft-thresholding operator $\Pi_\beta(X) = U\pi_\beta(\Sigma)V^T$, where $U\Sigma V^T$ is the skinny SVD of $X$. The update steps for $C_1$ and $C_2$ followed:

$$C_2 = \pi_{\frac{\lambda}{\mu_2}}\left(J + \frac{\Lambda_2}{\mu_2}\right), \qquad C_2 = C_2 - \operatorname{diag}(C_2), \qquad C_1 = \Pi_{\frac{1}{\mu_3}}\left(J + \frac{\Lambda_3}{\mu_3}\right). \tag{C.3}$$

Lastly, the dual variables are updated using gradient ascend:

$$\Lambda_1 = \Lambda_1 + \mu_1(X - XJ), \qquad \Lambda_2 = \Lambda_2 + \mu_2(J - C_2), \qquad \Lambda_3 = \Lambda_3 + \mu_3(J - C_1). \tag{C.4}$$

The full steps are summarized in Algorithm 1, with an optional adaptive penalty step proposed by Lin et. al[11]. Note that we deliberately constrain the proportion of $\mu_1$, $\mu_2$ and $\mu_3$ such that the $\left[\mu_1 X^T X + (\mu_2 + \mu_3)I\right]^{-1}$ need to be computed only once at the beginning.

## C.2 ADMM for NoisyLRSSC

The ADMM version of NoisyLRSSC is very similar to Algorithm 1 in terms of its Lagrangian and update rule. Again, we introduce dummy variable $C_1, C_2$ and $J$ to form

$$\begin{aligned} \min_{C_1, C_2, J} \quad & \frac{1}{2}\|X - XJ\|_F^2 + \beta_1\|C_1\|_* + \beta_2\|C_2\|_1 \\ s.t. \quad & J = C_2 - \operatorname{diag}(C_2), \quad J = C_1. \end{aligned} \tag{C.5}$$

Its Augmented Lagrangian is

$$\mathcal{L} = \|C_1\|_* + \lambda\|C_2\|_1 + \frac{1}{2}\|X - XJ\|_F^2 + \frac{\mu_2}{2}\|J - C_2 + \mathrm{diag}(C_2)\|_F^2$$
$$+ \frac{\mu_3}{2}\|J - C_1\|_F^2 + tr(\Lambda_2^T(J - C_2 + \mathrm{diag}(C_2))) + tr(\Lambda_3^T(J - C_1)),$$

and update rules are:

$$J = \left[X^TX + (\mu_2 + \mu_3)I\right]^{-1}\left[X^TX + \mu_2C_2 + \mu_3C_1 - \Lambda_2 - \Lambda_3\right] \tag{C.6}$$

$$C_2 = \pi_{\frac{\beta_2}{\mu_2}}\left(J + \frac{\Lambda_2}{\mu_2}\right), \qquad C_2 = C_2 - \mathrm{diag}(C_2), \qquad C_1 = \Pi_{\frac{\beta_1}{\mu_3}}\left(J + \frac{\Lambda_3}{\mu_3}\right). \tag{C.7}$$

Update rules for $\Lambda_2$ and $\Lambda_3$ are the same as in (C.4). Note that the adaptive penalty scheme also works for NoisyLRSSC but as there is a fixed parameter in front of $X^TX$ in (C.6) now, we will need to recompute the matrix inversion every time $\mu_2, \mu_3$ get updated.

### C.3 Convergence guarantee

Note that the general ADMM form is

$$\min_{x,z} \; f(x) + g(z)$$
$$s.t. \quad Ax + Bz = c. \tag{C.8}$$

In our case, $x = J$, $z = [C_1, C_2]$, $f(x) = \frac{1}{2}\|X - XJ\|_F^2$, $g(z) = \beta_1\|C_1\|_* + \beta_2\|C_2\|_1$ and constraints can be combined into a single linear equation after vectorizing $J$ and $[C_1, C_2]$. Verify that $f(x)$ and $g(z)$ are both closed, proper and convex and the unaugmented Lagrangian has a saddle point, then the convergence guarantee follows directly from Section 3.2 in [3].

Note that the reason we can group $C_1$ and $C_2$ is because the update steps of $C_1$ and $C_2$ are concurrent and do not depends on each other (see (C.3) and (C.7) and verify). This trick is important as the convergence guarantee of the three-variable alternating direction method is still an open question.

## D   Additional experimental results

### D.1   Numerical Simulation

**Exp1: Disjoint 11 Subspaces Experiment**

Randomly generate 11 subspaces of dimension 10 from $\mathbb{R}^{50}$. 50 unit length random samples are drawn from each subspace and we concatenate into a $50 \times 550$ data matrix. Besides what is shown in the main text, we provide a qualitative illustration of the separation-sparsity trade-off in Figure 1.

**Exp2: when exact SEP is not possible**

In this experiment, we randomly generate 10 subspaces of rank 3 from a 10 dimensional subspace, each sampled 15 data points. All data points are embedded to the ambient space of dimension 50.

This is to illustrate the case when perfect SEP is not possible for any $\lambda$. In other word, the smallest few singular values of the normalized Laplacian matrix is not exactly 0. Hence we will rely on heuristics such as Spectral Gap and Spectral Gap Ratio to tell how many subspaces there are and hopefully spectral clustering will return a good clustering. Figure 2 gives an qualitative illustration how the spectral gap emerges as $\lambda$ increases. Figure 3 shows quantitatively the same thing with the actual values of the two heuristics changes. Clearly, model selection is much easier in the SSC-side comparing to the LRR side, when SEP is the main issue (see the comparison in Figure 4).

Figure 1: Qualitative illustration of the 11 Subspace Experiment. From left to right, top to bottom: $\lambda = [0, 0.05, 1, 1e4]$, corresponding RelViolation is [3.4, 1.25, 0.06, 0.03] and Gini Index is [0.41, 0.56, 0.74, 0.79]

Figure 2: Last 50 Singular values of the normalized Laplacian in Exp2. See how the spectral gap emerges and become larger as $\lambda$ increases.

Figure 3: Spectral Gap and Spectral Gap Ratio for Exp2. When perfect SEP is not possible, model selection is easier on the SSC side, but the optimal spot is still somewhere between LRR and SSC.

Figure 4: Illustration of representation matrices. Left: $\lambda = 0$, Right: $\lambda = 1e4$. While it is still not SEP, there is significant improvement in separation.

**Exp3: Independent-Skewed data distribution**

Assume ambient dimension $n = 50$, 3 subspaces. The second and the third 3-d subspaces are generated randomly, each sampled 15 points. The first subspace is a 6-d subspace spanned by two random 3-d subspaces. 15 data points are randomly generated from each of the two spanning 3-d subspaces and only 3 data points are randomly taken from the spanned 6-D subspace two glue them together.

As a indication of model selection, the spectral gap and spectral ratio for all $\lambda$ is shown in Figure 5. While all experiments return clearly defined three disjoint components (smallest three singular values equal to 0 for all $\lambda$), the LRR side gives the largest margin of three subspaces (when $\lambda = 0$, the result gives the largest 4th smallest singular value). This illustrates that when Skewed-Data-Distribution is the main issue, LRR side is better than SSC side. This can be qualitatively seen in Figure 6

**Exp4: Disjoint-Skewed data distribution**

In this experiment, we illustrate the situation when subspaces are not independent and one of them has skewed distribution, hence both LRR and SSC are likely to to encounter problems. The setup

Figure 5: Spectral Gap and Spectral Gap Ratio for Exp3. The independent subspaces have no separation problem, SEP holds for all $\lambda$. Note that due to the skewed data distribution, the spectral gap gets quite really small at the SSC side.

Figure 6: Illustration of representation matrices. Left: $\lambda = 0$, Right: $\lambda = 1e4$. The 3 diagonal block is clear on the LRR side, while on the SSC side, it appear to be more like 4 blocks plus some noise.

is the same as the 6 Subspace experiment except the first two subspaces are combined into a 20-dimensional subspace moreover 10 more random points are sampled from the spanned subspace. Indeed, as Figure 2 and 3 in the main paper suggest, taking $\lambda$ somewhere in the middle gives the largest spectral gap and spectral gap ratio, which indicates with large margin that the correct model is a 5 Subspace Model.

In addition to that, we add Figure 7 here to illustrate the ranges of $\lambda$ where two heuristics give correct model selection. It appears that "spectral gap" suggests a wrong model for all $\lambda$ despite the fact that the $5^{th}$ "spectral gap" enlarges as $\lambda$ increase. On the other hand, the "spectral gap ratio" reverted its wrong model selection at the LRR side quickly as $\lambda$ increases and reaches maximum margin in the blue region (around $\lambda = 0.5$). This seems to imply that "spectral gap ratio" is a better heuristic in the case when one or more subspaces are not well-represented.

### D.2 Real Experiments on Hopkins155

To complement the numerical experiments, we also run our NoisyLRSSC on the Hopkins155 motion segmentation dataset[16]. The dataset contains 155 short video sequence with temporal trajectories

Figure 7: Illustration of model selection with spectral gap (left) and spectral gap ratio (right) heuristic. The highest point of each curve corresponds to the inferred number of subspaces in the data. We know the true number of subspace is 5.

Figure 8: Snapshots of Hopkins155 motion segmentation data set.

of the 2D coordinates of the feature points summarizing in a data matrix. The task is to unsupervisedly cluster the given trajectories into blocks such that each block corresponds to one rigid moving objects. The motion can be 3D translation, rotation or combination of translation and rotation. Ground truth is given together with the data so evaluation is simply by the misclassification rate. A few snapshots of the dataset is given in Figure 8.

### D.2.1 Why subspace clustering?

Subspace clustering is applicable here because collections of feature trajectories on a rigid body captured by a moving affine camera can be factorized into camera motion matrix and a structure matrix as follows

$$
X = \begin{pmatrix} x_{11} & ... & x_{1n} \\ ... & ... & ... \\ x_{m1} & ... & x_{mn} \end{pmatrix} = \begin{pmatrix} M_1 \\ ... \\ M_m \end{pmatrix} \begin{pmatrix} S_1 & ... & S_n \end{pmatrix},
$$

where $M_i \in \mathbb{R}^{2 \times 4}$ is a the camera projection matrix from 3D homogeneous coordinates to 2D image coordinates and $S_j \in \mathbb{R}^4$ is one feature points in 3D with 1 added at the back to form the homogeneous coordinates. Therefore, the inner dimension of the matrix multiplication ensures that all column vectors of $X$ lies in a 4 dimensional subspace (see [10, Chapter 18] for details).

Depending on the types of motion, and potential projective distortion of the image (real camera is never perfectly affine) the subspace may be less than rank 4 (degenerate motion) or only approximately rank 4.

Figure 9: Average misclassification rates vs. $\lambda$.

### D.2.2 Methods

We run the ADMM version of the NoisyLRSSC (C.5) using the same parameter scheme (but with different values) proposed in [9] for running Hopkins155. Specifically, we rescaled the original problem into:

$$\min_{C_1, C_2, J} \frac{\alpha}{2} \|X - XJ\|_F^2 + \alpha\beta_1\|C_1\|_* + \alpha\beta_2\|C_2\|_1$$
$$s.t. \quad J = C_2 - \text{diag}(C_2), \quad J = C_1,$$

and set

$$\alpha = \frac{\alpha_z}{\mu_z}, \qquad \beta_1 = \frac{1}{1+\lambda}, \qquad \beta_2 = \frac{\lambda}{1+\lambda}.$$

with $\alpha_z = 15000^3$, and

$$\mu_z = \min_i \max_{i \neq j} \langle x_i, x_j \rangle.$$

Numerical parameters in the Lagrangian are set to $\mu_2 = \mu_3 = 0.1\alpha$. Note that we have a simple adaptive parameter that remains constant for each data sequence.

Also note that we do not intend to tune the parameters to its optimal and outperform the state-of-the-art. This is just a minimal set of experiments on the real data to justify how the combinations of the two objectives may be useful when all other factors are equal.

### D.2.3 Results

Figure 9 plots how average misclassification rate changes with $\lambda$. While it is not clear on the two-motion sequences, the advantage of LRSSC is drastic on three motions.

To see it more clearly, we plot the RelViolation, Gini index and misclassification of all sequence for all $\lambda$ in Figure 11, Figure 12 and Figure 10 respectively. From Figure 11 and 12, we can tell that the shape is well predicted by our theorem and simulation. Since a correct clustering depends on both inter-class separation and intra-class connections, it is understandable that we observe the phenomena in Figure 10 that some sequences attain zero misclassification on the LRR side, some on the SSC side, and to our delight, some reaches the minimum misclassification rate somewhere in between.

Figure 10: Misclassification rate of the 155 data sequence against $\lambda$. Black regions refer to perfect clustering, and white regions stand for errors.

Figure 11: RelViolation of representation matrix $C$ the 155 data sequence against $\lambda$. Black regions refer to zero RelViolation (namely, SEP), and white regions stand for large violation of SEP.

Figure 12: GiniIndex of representation matrix $C$ the 155 data sequence againt $\lambda$. Darker regions represents denser intra-class connections, lighter region means that the connections are sparser.

Figure 13: The illustration of dual direction and its geometric meaning (figure extracted from [14]).

### D.2.4  Comparison to SSC results in [9]

After carefully studying the released SSC code that generates Table 5 in [9], we realized that they use two post processing steps on the representation matrix $C$ before constructing affinity matrix $|C| + |C^T|$ for spectral clustering. First, they use a thresholding step to keep only the largest non-zero entries that sum to 70% of the $\ell_1$ norm of each column. Secondly, there is a normalization step that scales the largest entry in each column to one (and the rest accordingly). The results with 4.4% and 1.95% misclassification rates for respectively 3-motion and 2-motion sequences essentially refer to the results with postprocessing.

Without postprocessing, the results we get are 5.67% for 3-motions and 1.91% for 2-motions. Due to the different implementation of the numerical algorithms (in stopping conditions and etc), we are unable to reproduce the same results on the SSC end (when $\lambda$ is large) with the same set of weighting factor, but we managed to make the results comparable (slightly better) with a different set of weighting even without any post-processing steps. Moreover, when we choose $\lambda$ such that we have a meaningful combination of $\ell_1$ norm and nuclear norm regularization, the 3-motion misclassification rate goes down to 3%.

Since the Hopkins155 dataset is approaching saturation, it is not our point to conclude that a few percentage of improvement is statistically meaningful, since one single failure case that has 40% of misclassification will already raise the overall misclassification rate by 1.5%. Nevertheless, we are delighted to see LRSSC in its generic form performs in a comparable level as other state-of-the-art algorithms.

## E  Discussions and bounds of minimax subspace incoherence property

In this section, we will explain the notion of minimax subspace incoherence property here (Definition 3) and highlight the difference between the new definition and the subspace incoherence property in [14].

### E.1  Non-uniqueness of the dual directions

Since the concept critically depends on the normalized dual direction matrix (Definition 2). That is what we we start with. $V(X)$ is essentially an optimal solution to the dual problem of LRSSC with data $X$. When $\lambda = \infty$, namely, in SSC's case, the dual problem is an LP, hence its solution may be obtained geometrically on the vertices of the dual polytope in a column-by-column fashion. This is illustrated in Figure 13, where the dual direction of data point $x_i^{(\ell)}$ is obtained from its low-dimensional representation $y$. Note that $x_i^{(\ell)} = Uy$ for some orthonormal basis $U$ of $\mathcal{S}_\ell$. Other data points in $\mathcal{S}_\ell$ can be similarly represented as $X_{-i}^{(\ell)} = UA$. Note that the reduced dimensional primal constraint $y = Ac$ is equivalent to the original $x_i^{(\ell)} = X_{-i}^{(\ell)}c$. Dual point of the reduced dimensional dual problem is obtained and denoted as $\lambda(A, y)$ and the dual direction $v_i^{(\ell)}$ corresponding to $x_i^{(\ell)}$ is

hence defined as the embedding of the low-dimensional dual point $\lambda(y, A)$ to the ambient space via

$$v_i^{(\ell)} = U\lambda(y, A)/\|\lambda(y, A)\|.$$

In the general LRSSC case it is an SDP, hence there is no simple geometric illustration of where the optimal dual variable will be. In addition, since nuclear norm cannot be separated into column by column optimization, the dual variable is defined as a matrix. Nevertheless, the key idea is the same. We may still represent the data in the low-dimensional space and obtain a dual matrix $V^*(X^{(\ell)})$ where all columns of which are within the subspace of $X^{(\ell)}$.

The key observation here in this paper is that the dual matrix constructed in this way is NOT the only optimal dual matrix. Essentially, in the ambient space, we may add any arbitrary matrix $V^\perp(X^{(\ell)})$ to $V^*(X^{(\ell)})$ as long as each column of $V^\perp(X^{(\ell)})$ belongs to the orthogonal complement of $\mathcal{S}_\ell$. The so-called normalized dual matrix set is just the collection of all possible dual matrices with each column's central component in $\mathcal{S}_\ell$ normalized to 1.

### E.2 The advantages of the minimax subspace incoherence property

The minimax subspace incoherence (Definition 3) is simply defined as the minimum subspace incoherence over all possible dual matrix

$$V^\perp(X^{(\ell)}) = V^*(X^{(\ell)}) + V^\perp(X^{(\ell)}).$$

It differs from the original definition in [14] in that [14] takes $V^\perp(X^{(\ell)}) = 0$. There is two effects of using a non-zero $V^\perp(X^{(\ell)})$. First, the magnitude of each column will be larger. This is undesirable since we would like $\|[V(X^{(\ell)})]^T x\|_\infty$ to be as small as possible. Another effect is on the angles between each column of $V(X^{(\ell)})$ and $x$. This is something desirable since we may choose a direction such that the angles approach $\pi/2$ for all $x$. This is the property we will leverage upon in the proof of Example 1 and 2, which demonstrate that in many cases, using a non-zero $V^\perp(X^{(\ell)})$ leads to substantially smaller incoherence $\mu$.

#### E.2.1 Proof of Example 1(Independent subspace)

We claim in Example 1 that $\mu = 0$ when subspaces are independent without detailed justification. Here we provide the proof and an illustration. By definition of independent subspaces, $\dim(\mathcal{S}_1 \oplus ... \oplus \mathcal{S}_L) = \sum_{\ell=1,...,L} \dim(\mathcal{S}_\ell) \leq n$ where $n$ is the ambient dimension. Then for data point $x$ in $\mathcal{S}_i$, we may choose a corresponding dual vector $\nu = \nu^* + \nu^\perp$ such that

$$\nu \in \text{Null}(\mathcal{S}_1 \oplus ... \oplus \mathcal{S}_{i-1} \oplus \mathcal{S}_{i+1} \oplus ... \oplus \mathcal{S}_L)).$$

The nullspace is of dimension larger than 1 if we remove any $\mathcal{S}_i$, so we can always construct such $\nu$ (with potentially very large $\nu^\perp$). Then by definition of subspace incoherence $\nu = 0$ is proven. The construction is illustrated in Figure 14.

#### E.2.2 Proof of Example 2 (Random except 1)

Recall that the setup is $L$ disjoint 1-dimensional subspaces in $\mathbb{R}^n$ ($L > n$). $\mathcal{S}_1, ..., \mathcal{S}_{L-1}$ subspaces are randomly drawn. $\mathcal{S}_L$ is chosen such that its angle to one of the $L - 1$ subspace, say $\mathcal{S}_1$, is $\pi/6$. There is at least one samples in each subspace, so $N \geq L$. Our claim is that

**Proposition E.2.1.** *Assume the above problem setup and Definition 3, then with probability at least* $1 - 2L/N^3$

$$\mu \leq 2\sqrt{\frac{6\log(L)}{n}}.$$

*Proof.* The proof is simple. For $x_i \in \mathcal{S}_\ell$ with $\ell = 2, ..., L - 1$, we simply choose $\nu_i = \nu_i^*$. Note that $\nu_i^*$ is uniformly distributed, so by Lemma B.3.1 and union bound, the maximum of $|\langle x, \nu_i \rangle|$ is upper bounded by $2\sqrt{\frac{6\log(N)}{n}}$ with probability at least $1 - \frac{2(L-2)^2}{N^{12}}$. Then we only need to consider $\nu_i$ in $\mathcal{S}_1$ and $\mathcal{S}_L$, denoted by $\nu_1$ and $\nu_L$. We may randomly choose any $\nu_1 = \nu_1^* + \nu_1^\perp$ obeying $\nu_1 \perp \mathcal{S}_L$ and similarly $\nu_L \perp \mathcal{S}_1$.

Figure 14: Illlustration of how dual vector $\nu$ can be constructed to get minimax subspace incoherence $\mu = 0$ under independent subspace assumption. Note that we can always find a $\nu$ perpendicular to the span to the remaining subspaces no matter how closely affiliated the subspaces are.

By the assumption that $\angle(\mathcal{S}_1, \mathcal{S}_L) = \pi/6$,

$$\|\nu_1\| = \|\nu_L\| = \frac{1}{\sin(\pi/6)} = 2.$$

Also note that they are considered a fixed vector w.r.t. all random data samples in $\mathcal{S}_2, .., \mathcal{S}_L$, so the maximum inner product is $2\sqrt{\frac{6\log(N)}{n}}$, summing up the failure probability for the remaining $2L - 2$ cases, we get

$$\mu \leq 2\sqrt{\frac{6\log(N)}{n}} \quad \text{with probability } 1 - \frac{2L-2}{N^3} - \frac{2(L-2)^2}{N^{12}} > 1 - \frac{2L}{N^3}.$$

$\square$

### E.3  "Sufficiently Independent": Take-$K$-out-Independence

We mention in the Example 3 of the main paper that as long as the subspaces are "sufficiently Independent", subspace incoherence $\mu$ will be significantly smaller under our minimax definition than the original subspace incoherence definition in [14]. In this section, we formalize our claim with by introducing the novel Take-$K$-out-Independence condition and providing a bound of incoherence $\mu$ under both deterministic and randomized model.

**Definition E.3.1** (Take-$K$-out-Independence). *Suppose there are $L$ disjoint subspaces, if we take out any $K$ subspaces from it, then it becomes independent, then we say these $L$ subspaces obey "Take-$K$-Out-Independence".*

**Definition E.3.2** (Take-$K$-out-Angle). *Correspondingly, let the indices of $K$ subspaces taken out be $\mathcal{K}$ and the remaining subspaces indices be $\mathcal{K}^c := \{1, ..., L\}/\mathcal{K}$, furthermore, denote each $\binom{L}{K}$ experiment with index $i$ such that $\mathcal{K}_i$ and $\mathcal{K}_i^c$ represents the particular indices set for experiment $i$ and $\mathcal{A}_{(i)}^{-\ell} := \mathrm{span}(\mathcal{S}_k | k \in \mathcal{K}_i^c/\ell)$. Then we may we define "Take-$K$-out-Angle" as*

$$\theta = \arcsin\left[\min_i \min_{\ell \in \mathcal{K}_i^c} \min_{\{j|x_j \in X^{(\ell)}\}} \|\mathrm{Proj}_{\mathrm{Null}(\mathcal{A}_{(i)}^{-\ell})}\left(\nu_j^*\right)\|\right],$$

*where $\nu_j^* \in \mathcal{S}$ is the central dual vector corresponding to $x_j$. $\mathrm{Proj}_{\mathcal{A}}$ is the Euclidean projection to subspace $\mathcal{A}$ and $\mathrm{Null}(\mathcal{A})$ gives the null space of subspace $\mathcal{A}$. Note that if $\mathcal{S}_1, ..., \mathcal{S}_L$ obeys Take-$K$-Out-Independence, then $\dim[\mathrm{Null}(\mathcal{A}_{(i)}^{-\ell})] \geq 1$.*

**Proposition E.3.1** ($\mu$ bound for deterministic Take-$K$-Out-Independent subspaces). *If $L$ subspaces obey Take-$K$-out-Independence with Take-$K$-out-Angle $\theta$, then the minimax subspace incoherence property in Definition 3 is upper bounded with*

$$\mu \leq \frac{K}{(L-1)\sin\theta}. \tag{E.1}$$

**Example E.3.1** (Trivial cases). *If subspaces are independent, $K = 0$, then $\mu = 0$. If subspaces are "near independent" by $K$ the smaller $K$ is, the better the bound. There might be a range of $K$ under which this bound of $\mu$ is meaningful.*

*Proof of Proposition E.3.1.* We prove the inequality by constructing a normalized dual matrix $V^{(\ell)}$ for each subspace $\mathcal{S}_\ell$. For all $\binom{L-1}{K}$ cases that include $\mathcal{S}_\ell$, the $L - K$ remaining subspaces are independent. So we may take $V^{(\ell)} = [V^{(\ell)}]^* + [V^{(\ell)}]^\perp$ such that $V^{(\ell)}$ is orthogonal to all other $L - K - 1$ subspaces. By the angle assumption, each column is bounded above by $\frac{1}{\sin\theta}$ .

It follows that, $\|[V^{(\ell)}]^T X^{(k)}\|_\infty$ is 0 if $\mathcal{S}_k$ is not taken out. Otherwise,

$$\|[V^{(\ell)}]^T X^{(k)}\|_\infty \leq \max_i \|V_i^{(\ell)}\| \leq \frac{1}{\sin\theta}.$$

Now if we take $V^{(\ell)}$ to be the average of all $N = \binom{L-1}{K}$ cases, for each $k$,

$$[V^{(\ell)}]^T X^{(k)} = \frac{1}{N} \sum_{i=1}^{N} [V^{(\ell)}]_i^T X^{(k)},$$

Note that in only $\binom{L-2}{K-1}$ cases out of all $N$, $[V^{(\ell)}]_i^T X^{(k)}$ is non-zero (when $k$ is chosen to be one of the $K$ taken out). With this observation,

$$\|[V^{(\ell)}]^T X^{(k)}\|_\infty \leq \binom{L-2}{K-1} / \binom{L-1}{K} (\frac{1}{\sin\theta}) = \frac{K}{(L-1)\sin\theta}.$$

Verify that $V^{(\ell)}$ constructed this way can still be decomposed into unit column $[V^{(\ell)}]^*$ and $[V^{(\ell)}]^\perp$ orthogonal to $\mathcal{S}_\ell$, so it is a valid normalized dual matrix for $X^\ell$.

By construct such $V^{(\ell)}$ for each subspace $\mathcal{S}_\ell$, we complete the proof. $\qquad\square$

By assuming the data are randomly generated, we are able to obtain a much better bound. It uses a similar way of constructing dual variables as above, but is in a sense adaptive to the size of $\sin\theta$ in each trial. Moreover, to get rid of the $\sin\theta$ all together, we derived a bound of the probability that $\sin\theta$ is greater than any positive value. The result is essentially a lower bound of the area of spherical cap (the opposite of Lemma B.3.1) and is interesting in its own light. To not get distracted, we state the results in Section F.3 focus here on results for random "Take-$K$-Out-Independent" subspaces.

**Proposition E.3.2** ($\mu$ bound for random Take-$K$-Out-Independent subspaces). *Suppose ambient dimension is $n$, if $L$ subspaces and a total of $N$ data points are randomly generated, furthermore they obeys "Take-$K$-Out-Independence" (e.g., sufficiently each subspace is rank-$d$ and $n < Ld < n + Kd$) and $M := \binom{L-1}{K}$, then with probability larger than $1 - 3/N$, the minimax subspace incoherence*

$$\mu \leq \frac{K\sqrt{6\log N}}{\alpha(L-1)} + \sqrt{\frac{6\log N}{n}} \left[1 - (1 - \delta(\alpha, n))e^{-3\alpha^2/2}\right] + \frac{\sqrt{12}\log N}{\sqrt{nM}}, \tag{E.2}$$

*for any $\alpha > 0$. Small residual*

$$\delta(\alpha, n) < \begin{cases} \frac{e}{2(n+1)!} + \frac{\alpha^2}{n}, & when\ 0 < \alpha < \sqrt{\frac{2}{3}}; \\ \frac{e}{2(n+1)!} + \frac{\alpha^4}{n}, & otherwise. \end{cases}$$

*Moreover, when $\alpha > \Theta(\sqrt{\frac{\log N}{n}})$ is sufficiently large or if $\alpha < o(e^{-n})$, then for the same probability, $\mu$ satisfies respectively*

$$\mu \leq \sqrt{\frac{6\log N}{n}}, \qquad\qquad \mu < \frac{K\sqrt{6\log N}}{\alpha(L-1)},$$

*for probability larger then $1 - 3/N$.*

**Example E.3.2** ($n + 1$ i.i.d 1D subspaces). In this case, $K = 1$, $L = n + 1$, $M = n$, suppose $n$ is large such that $\log N/n \ll \sqrt{\log N/n}$, we may take $\alpha = 0.1$ and get

$$\mu < \frac{24.5 \log N}{n - 1} + 0.015 \sqrt{\frac{6 \log N}{n}} + \frac{\sqrt{12} \log N}{n} < 0.03 \frac{6 \log N}{\sqrt{n}}.$$

This is more than 20 times smaller than the bound in Lemma 3.

**Example E.3.3** ($\lfloor n/d + K \rfloor$ i.i.d. rank-$d$ subspaces). This is a generalization of previous example. $L = \lfloor n/d + K \rfloor = \lfloor n/d \rfloor + K$, $M = \binom{L-1}{K}$. As $K$ and $d$ increases,

As $Kd$ increases, the first term of (E.2) gets larger. Orderly speaking, whenever $Kd = o(\sqrt{n})$, the bound here is better than Lemma 3.

We may verify that this indeed happens by checking $M = \binom{\lfloor n/d \rfloor + K - 1}{K}$ increases monotonically w.r.t. the increase of $K$ and the decrease of $d$ in the range of $Kd = o(\sqrt{n})$. The smallest $M$ occur at $K = 1$ and $d = \lfloor \sqrt{n} \rfloor$, where $M = \lfloor \sqrt{n} \rfloor$. This implies that the third term of (E.2) is small compare to the first term in all interesting cases.

**Example E.3.4** (Independent subspaces). Note that when subspaces are independent, $K = 0$, then we may choose arbitrarily small $\alpha$ so that naturally the upper bound approaches 0.

**Example E.3.5** (Large $K$). When $K$ is large, meaning that it is by no means near independent, then we may choose $\alpha = \infty$, then the bound is the same as that in Lemma 3.

*Proof of Proposition E.3.2.* We prove the inequality by constructing a normalized dual vector $\nu$ for data $x$ in subspace data $X_1$ and then take the minimum over the normalized dual vector for all data points. To find the inner product of $\nu$ against all other $y \in X_{\ell \neq 1}$, again we pick one such $y$ and consider $\langle \nu, y \rangle$ only.

Now consider the procedure of "Take-$K$-Out" experiments, there are $M = \binom{L-1}{K}$ experiments with $\mathcal{S}_1$ not taken out. Among them, there are respectively $M_1 = \binom{L-2}{K-1}$ and $M_2 = \binom{L-2}{K}$ trials when one particular $y$ is inside the $K$ and inside the $L - 1 - K$ remaining subspaces. Conveniently,

$$\frac{M_1}{M} = \frac{K}{L - 1}, \qquad \frac{M_2}{M} = \frac{L - K - 1}{L - 1}, \qquad M_1 + M_2 = M.$$

Let the $\nu = \nu^* + \nu^\perp$, where $\nu^* \in \mathcal{S}_1$ is the central dual vector with unit norm and $\nu^\perp \in \mathcal{S}_1^\perp$. Here we are going to construct $\nu_1, ..., \nu_M$ for each and every experiments then derive a bound for $|\langle \nu, y \rangle|$ with

$$\nu = \frac{1}{M} \sum_{i=1}^{M} \nu_i = \nu^* + \sum_{i=1}^{M} \nu_i^\perp.$$

For each experiment, the $L - K$ subspaces are independent, so by taking out $\mathcal{S}_1$, the span of the remaining $L - K - 1$ subspaces do not cover the full ambient space, in other word, there is a null space $\mathrm{Null}(A_i)$ for the data matrix $A_i$ containing all samples in the $L - K - 1$ subspaces. Project $\nu^*$ to $\mathrm{Null}(A_i)$ and normalize it to unit vector $n_i$. Note that $n_i$ is the normal vector of the hyperplane $\mathrm{span}(A_i)$ that is closest to $\nu^*$.

Then we can construct $\nu_i$ by considering only $\nu_i^\perp$ in the 2-$D$ plane spanned by $\nu^*$ and $n_i$. Because it is planar, we can use simple trigonometry to express $\langle y, \nu_i \rangle$ analytically. The procedure is illustrated in Figure 15. Note that $\theta_i$ is the angle between $\nu^*$ and the intersecting line $\mathcal{L} =: \mathrm{span}(\nu^*, n_i) \cap \mathrm{span}(A_i)$ and $\phi_i$ is the angle between $\nu^*$ and $\nu_i$. The angle $\phi_i$ characterizes how much we want to push $\nu_i$ from $\nu^*$ towards $n_i$.

Note that if we project $\nu^*$ and $\nu_i^\perp$ to the intersecting line, the inner product can be Now we consider the inner product

$$\langle y, \nu_i \rangle = \langle y, \nu^* \rangle + \langle y, \nu_i^\perp \rangle.$$

When $y$ is inside the $K$ subspaces taken out, there is nothing we can do to simplify the form. Otherwise, we can express it by $\nu^*$ alone. The mechanism $\nu_i^\perp$ reduces the inner product is essentially reducing the magnitude of $\nu$'s projection to the line $\mathcal{L}$ by a factor (see Figure 15). Algebraically, we have

$$\langle y, \nu_i \rangle = \langle y, \mathrm{Proj}_{\mathcal{L}}(\nu^* + \nu_i^\perp) \rangle = \langle y, \mathrm{Proj}_{\mathcal{L}}(\nu^*) \rangle + \langle y, \mathrm{Proj}_{\mathcal{L}}(\nu_i^\perp) \rangle.$$

Figure 15: Illustration of how $\nu_i$ is constructed with $\nu_i^*$ and $\phi_i$ in the plane spanned by $n_i$ and $\nu_i^*$. Note that $i$ is the index of this experiment where $\mathcal{S}_4$ is taken out. $\mathcal{S}_1$, $\mathcal{S}_2$ and $\mathcal{S}_3$ are hereby independent. Also note that we can tune $\phi_i$ to obtain the optimal incoherence value.

Since the direction $\nu_i^\perp$ is always chosen to reduce this inner product,

$$\begin{aligned}
\langle y, \nu_i \rangle &= (\|\mathrm{Proj}_{\mathcal{L}}(\nu^*)\| - \|\mathrm{Proj}_{\mathcal{L}}(\nu_i^\perp)\|)\langle y, \mathrm{Proj}_{\mathcal{L}}(\nu^*)/\|\mathrm{Proj}_{\mathcal{L}}(\nu^*)\|\rangle \\
&= (\cos\theta_i - \sin\theta_i \tan\phi_i)\langle y, \mathrm{Proj}_{\mathcal{L}}(\nu^*)/\cos\theta_i\rangle = (1 - \tan\theta_i \tan\phi_i)\langle y, \mathrm{Proj}_{\mathcal{L}}(\nu^*)\rangle \\
&= (1 - \tan\theta_i \tan\phi_i)\langle y, \nu^*\rangle.
\end{aligned}$$

Moreover, we choose the value of $\phi_i = \phi(\theta_i, L, K)$ defined in the following manner

$$\phi(\theta, L, K) = \begin{cases} \frac{\pi}{2} - \theta, & \text{if } \sin\theta \geq \frac{\alpha}{\sqrt{n}}\,; \\ 0, & \text{otherwise.} \end{cases} \tag{E.3}$$

Note that in the first case, $\phi_i = \pi/2 - \theta_i$, then

$$\langle y, \nu_i \rangle = \langle y, \nu^*\rangle + \langle y, \nu_i^\perp\rangle = \langle y, \nu^*\rangle + \langle y, \frac{1}{\sin\theta_i}n_i - \nu_*\rangle = \frac{1}{\sin\theta_i}\langle y, n_i\rangle.$$

In the second case, $\tan\phi_i = 0$. For simplicity, denote the event that $\sin\theta_i \geq \frac{\alpha}{\sqrt{n}}$ to be $E_1$ and let $E_2$ to be its complement, then we have

$$\langle y, \nu_i \rangle = \begin{cases} \langle y, \nu^*\rangle, & \text{if } E_2; \\ \frac{1}{\sin\theta_i}\langle y, n_i\rangle, & \text{if } E_1 \text{ AND } y \in \{K\}; \\ 0, & \text{Otherwise.} \end{cases}$$

We count the total number of $E_1$ among all $M$ experiments and obtain the empirical probability

$$\hat{p} = \frac{1}{M}\sum_{i=1,\ldots,M} \mathbb{1}\left\{\sin\theta_i \geq \frac{\alpha}{\sqrt{n}}\right\}.$$

Note that $\mathbb{1}\{\cdot\}$ is the indicator function. Denote $\hat{p}_1$ and $\hat{p}_2$ to be the corresponding empirical probability of $E_1$ in the $M_1$ cases when $y \in \{K\}$ and in the $M_2$ cases when $y \notin \{K\}$ respectively. Verify that

$$\hat{p}_1 M_1 + \hat{p}_2 M_2 = \hat{p}M.$$

Also note that the empirical probability of $E_2$ is exactly $1 - \hat{p}$. Then it follows that

$$
\begin{aligned}
\langle y, \nu \rangle =& \langle y, \frac{1}{M} \sum_{i=1}^{M} \nu_i \rangle = \frac{1}{M} \left[ \sum_{\{i | y \in \{K\}\}} \langle y, \nu_i \rangle + \sum_{\{i | y \in \{K\}^c\}} \langle y, \nu_i \rangle \right] \\
=& \frac{1}{M} \left[ \sum_{\{i | y \in \{K\} \cap E_1\}} \frac{1}{\sin \theta_i} \langle y, n_i \rangle + \sum_{\{i | y \in \{K\} \cap E_2\}} \langle y, \nu^* \rangle + \sum_{\{i | y \in \{K\}^c \cap E_2\}} \langle y, \nu^* \rangle \right] \\
=& \frac{p_1 M_1}{M} \left[ \frac{1}{p_1 M_1} \sum_{\{i | y \in \{K\} \cap E_1\}} \frac{1}{\sin \theta_i} \langle y, n_i \rangle \right] + (1 - \hat{p}) \langle y, \nu^* \rangle = \langle y, \tilde{\nu} \rangle
\end{aligned}
$$

where

$$
\tilde{\nu} = \frac{1}{M} \sum_{\{i | y \in \{K\} \cap E_1\}} \frac{n_i}{\sin \theta_i} + (1 - \hat{p}) \nu^*. \tag{E.4}
$$

To bound $|\langle y, \nu \rangle|$, we only need to bound $\|\tilde{\nu}\|$. By the definition of event $E_1$,

$$
\sin \theta_i > \frac{\alpha}{\sqrt{n}},
$$

then

$$
\|\tilde{\nu}\| \leq \frac{\sqrt{n} M_1 \hat{p}_1}{\alpha M} + 1 - \hat{p} = \frac{K \sqrt{n} \hat{p}_1}{\alpha (L - K - 1)} + 1 - \hat{p}
$$

Under the fully random assumption, $\tilde{\nu}$ is independent to $y$. This can be seen from (E.4) that $n_i$ and $\nu_*$ are both independent to the sampling of $y$ (since $y$ is among the data points of $K$ subspaces taken out). Thus we may apply Lemma B.3.1 to bound the inner product then use union bound to cover a total of less than $N^2$ number of events. With probability larger than $1 - \frac{2}{N}$, every event obeys

$$
|\langle y, \nu \rangle| \leq \frac{K \hat{p}_1 \sqrt{6 \log N}}{\alpha (L - 1)} + \frac{\sqrt{6 \log N}(1 - \hat{p})}{\sqrt{n}}. \tag{E.5}
$$

At this stage, we discuss three different cases of $\alpha$, corresponding to the three statements in Proposition E.3.2.

**(1) The general statement:** The general statement (E.2) by substituting the two empirical probability $\hat{p}_1$ and $\hat{p}$ by

$$
\hat{p}_1 \leq 1, \qquad\qquad \hat{p} \geq (1 - \delta(\alpha, n)) e^{-3\alpha^2/2} - \epsilon
$$

The first inequality is trivial. To prove the second, we consider $M$ i.i.d. Bernoulli experiments that get 1 if the event is $E_1$ and 0 otherwise. By Hoeffding's inequality, empirical expectation

$$
\hat{p} > p - \epsilon
$$

with probability larger than $1 - e^{-\epsilon^2 M}$. By Corollary F.3.1, we have $p > (1 - \delta(\alpha, n)) e^{-\frac{3\alpha^2}{2}}$. Also, we may choose $\epsilon = \sqrt{\frac{3 \log N}{M}}$ such that the failure probability over all $N^2$ events are less than $1/N$.

Substitute the bounds into (E.5) and combine all failure probabilities with union bound, we get

$$
\mu \leq \frac{K \sqrt{6 \log N}}{\alpha (L - 1)} + \frac{\sqrt{6 \log N} \left[ 1 - (1 - \delta(\alpha, n)) e^{-3\alpha^2/2} \right]}{\sqrt{n}} + \frac{\sqrt{12 \log N}}{\sqrt{nM}}. \tag{E.6}
$$

with probability larger than $1 - 3/N$. This gives us the general statement in Proposition E.3.2.

Now we will discuss two boundary cases of interest without using Hoeffding's inequality.

**(2) When $\alpha$ is large :** When $\alpha$ is sufficiently large, the last term in (E.6) can be removed. Denote the pdf of random inner product of Lemma F.3.1 as $f(x)$, then by definition

$$
p_\alpha = 2 \int_\alpha^\infty f(x) \mathrm{d}x.
$$

Naturally, there exists an $\tilde{\alpha}$ such that $p_{\tilde{\alpha}} = \frac{1}{MN^3}$ (in particular, by Lemma B.3.1, we may show $p_\alpha \le \frac{1}{MN^3}$ when $\alpha = \sqrt{\frac{6\log N + 2\log M}{n}}$. By union bound, the probability that $E_1$ does not occur in all $M$ events for all $N^2$ pairs $(x, \nu)$, is greater than $1 - 1/N$. So we may take $\hat{p} = 0$ and $\hat{p}_1 = 0$ in (E.5) and get directly the result

$$\mu \le \sqrt{\frac{6\log N}{n}}$$

with probability larger than $1 - 3/N$ for some sufficiently large $\alpha$.

**(3)When $\alpha$ goes to** $0$**:** Using a similar argument, when $\alpha$ is sufficiently small (typically smaller than $e^{-n}$), we can show that with probability larger than $1 - 1/N$, $E_2$ does not occur at all, hence $\hat{p} = 1$ and $\hat{p}_1 = 1$. Then from (E.5) directly, we may get

$$\mu \le \frac{K\sqrt{6\log N}}{\alpha(L-1)}$$

with probability larger than $1 - 3/N$. As $\alpha$ appear in the denominator, this bound is only meaningful when $K = 0$, which reflects the fact that $\mu = 0$ for independent subspace. The proof is now complete.

$\square$

# F    Other results and proofs

## F.1    Proof of Proposition 1 (LRR is dense)

For easy reference, we copy the statement of Proposition 1 here.

**Proposition F.1.1.** *When the subspaces are independent and $X$ is not full rank and the data points are randomly sampled from a unit sphere in each subspace, then the solution to LRR is class-wise dense, namely each diagonal block of the matrix $C$ is all non-zero.*

*Proof.* The proof is of two steps. First we prove that because the data samples are random, the shape interaction matrix $VV^T$ in Lemma 4 is a random projection to a rank-$d_\ell$ subspace in $\mathbb{R}^{N_\ell}$. Furthermore, each column is of a random direction in the subspace.

Second, we show that with probability 1, the standard bases are not orthogonal to these $N_\ell$ vectors inside the random subspace. The claim that $VV^T$ is dense can hence be deduced by observing that each entry is the inner product of a column or row[4] of $VV^T$ and a standard basis, which follows a continuous distribution. Therefore, the probability that any entries of $VV^T$ being exactly zero is negligible. $\square$

## F.2    Condition (2) in Theorem 1 is computational tractable

First note that $\mu(X^{(\ell)})$ can be computed by definition, which involves solving one quadratically constrained linear program (to get dual direction matrix $[V^{(\ell)}]^*$) then finding $\mu(X^{(\ell)})$ by solving the following linear program for each subspace

$$\min_{V^{(\ell)}} \; \|[V^{(\ell)}]^T \overline{X^{(\ell)}}\|_\infty \quad s.t. \quad \mathrm{Proj}_{\mathcal{S}_\ell} V^{(\ell)} = [V^{(\ell)}]^*,$$

where we use $\overline{X^{(\ell)}}$ to denote $[X^{(1)}, ..., X^{(\ell-1)}, X^{(\ell+1)}, ..., X^{(L)}]$.

To compute $\sigma_{d_\ell}(X_{-k}^{(\ell)})$, one needs to compute $N_\ell$ SVD of the $n \times (N_\ell - 1)$ matrix. The complexity can be further reduced by computing a close approximation of $\sigma_{d_\ell}(X_{-k}^{(\ell)})$. This can be done by finding the singular values of $X^{(\ell)}$ and use the following inequality

$$\sigma_{d_\ell}(X_{-k}^{(\ell)}) \ge \sigma_{d_\ell}(X^{(\ell)}) - 1.$$

This is a direct consequence of the SVD perturbation theory [15, Theorem 1].

Figure 16: Illustration of how Corollary F.3.1 approximates Lemma F.3.2 as $n$ increases for all values of $\alpha$.

## F.3 Lower bound of random inner product

**Lemma F.3.1** (pdf of inner product of random unit vectors[6]). *Let $u$, $v$ be random vectors uniformly distributed on the standard unit $n$-sphere and then the pdf of $z = \langle u, v \rangle$ is given as*

$$f_n(z) = \begin{cases} \frac{\Gamma(\frac{n+1}{2})}{\Gamma(\frac{n}{2})\sqrt{\pi}}\sqrt{1-z^2}^{n-2}, & for -1 < z < 1; \\ 0, & elsewhere, \end{cases} \tag{F.1}$$

*for $n = 1, 2, 3, ...$*

**Lemma F.3.2** (Lower bound of inner product of random unit vectors). *Suppose $x$ is independently sampled from unit $n$-sphere $S^{n-1}$. $y$ is a fixed vector. Then*

$$\Pr(|\langle x, y \rangle| > z_0\|y\|) > \left(1 - \frac{e}{2(n+1)!}\right)(1-z_0^2)^{\frac{3n}{2}}.$$

**Corollary F.3.1.** *A special case of interest is that when $z_0 = \frac{\alpha}{\sqrt{n}}$,*

$$\Pr\left(|\langle x, y \rangle| > \frac{\alpha}{\sqrt{n}}\|y\|\right) > (1 - \delta(\alpha, n))e^{-\frac{3\alpha^2}{2}}.$$

*where*

$$\delta(\alpha, n) < \begin{cases} \frac{e}{2(n+1)!} + \frac{\alpha^2}{n}, & when \ \alpha < \sqrt{\frac{2}{3}}; \\ \frac{e}{2(n+1)!} + \frac{\alpha^4}{n}, & otherwise. \end{cases}$$

The bound is tighter when $\alpha$ is small and when $n$ is large. A numerical comparison of the Corollary F.3.1 against the Lemma F.3.2 for different $\alpha$ and different $n$ are given in Figure 16.

*Proof of Corollary F.3.1.* This is a simple use of $(1 - \frac{1}{n})^n \to e^{-1}$ when $n$ is large. First note that when $x > 1$,

$$(1 - 1/x)^x > (1 - 1/x)(1 - 1/x)^{x-1} > (1 - 1/x)e^{-1}.$$

Then substitute $x = \frac{n}{\alpha^2}$ we get

$$(1 - \frac{\alpha^2}{n})^{\frac{n}{\alpha^2}} > (1 - \frac{\alpha^2}{n})e^{-1}.$$

Then we may simplify the result in Lemma F.3.2.

$$\Pr\left(|\langle x, y \rangle| > \frac{\alpha}{\sqrt{n}}\|y\|\right) > \left(1 - \frac{e}{2(n+1)!}\right)\left(1 - \frac{\alpha^2}{n}\right)^{\frac{3n}{2}}$$

$$= \left(1 - \frac{e}{2(n+1)!}\right)\left[\left(1 - \frac{\alpha^2}{n}\right)^{\frac{n}{\alpha^2}}\right]^{\frac{3\alpha^2}{2}}$$

$$> \left(1 - \frac{e}{2(n+1)!}\right)\left(1 - \frac{\alpha^2}{n}\right)^{\frac{3\alpha^2}{2}} e^{\frac{-3\alpha^2}{2}}$$

When $x \geq 1$, it holds that $(1 - \delta)^x > (1 - \delta x)$. Also, $(1 - a)(1 - b) > 1 - a - b$ when $a, b > 0$. So when $3\alpha^2/2 > 1$, or equivalently $\alpha > \sqrt{\frac{2}{3}}$

$$\left(1 - \frac{e}{2(n+1)!}\right)\left(1 - \frac{\alpha^2}{n}\right)^{\frac{3\alpha^2}{2}} > \left(1 - \frac{e}{2(n+1)!} - \frac{3\alpha^4}{2n}\right)$$

otherwise we may simply drop the exponent and get

$$\left(1 - \frac{e}{2(n+1)!}\right)\left(1 - \frac{\alpha^2}{n}\right)^{\frac{3\alpha^2}{2}} > \left(1 - \frac{e}{2(n+1)!} - \frac{\alpha^2}{n}\right)$$

$\square$

*Proof of Lemma F.3.2.* The probability $p_1$ of random inner product greater than $z_0$ is given by the following integral

$$p_1(z_0) = \frac{2\Gamma(\frac{n+1}{2})}{\Gamma(\frac{n}{2})\sqrt{\pi}} \int_{z_0}^1 \sqrt{1 - z^2}^{n-2} \, dz$$

$$= \frac{2\Gamma(\frac{n+1}{2})}{\Gamma(\frac{n}{2})\sqrt{\pi}} \int_{\arcsin z_0}^{\pi/2} \cos^{n-2}\theta \, d\sin\theta$$

$$= \frac{2\Gamma(\frac{n+1}{2})}{\Gamma(\frac{n}{2})\sqrt{\pi}} \int_{\arcsin z_0}^{\pi/2} \cos^{n-1}\theta \, d\theta$$

By the table of integral,

$$\int \cos^{n-1}\theta \, d\theta = -\frac{1}{n}\cos^n\theta \times {}_2F_1\left(\frac{n}{2}, \frac{1}{2}, \frac{n+1}{2}, \cos^2(\theta)\right) + C,$$

where ${}_2F_1[a, b; c; z]$ is the so-called Gauss's hypergeometric function, defined as follows:

$${}_2F_1(a, b; c; z) = \sum_{n=0}^{\infty} \frac{(a)_n (b)_n}{(c)_n} \frac{z^n}{n!},$$

where

$$(q)_n = \begin{cases} 1, & \text{if } n = 0; \\ q(q+1)...(q+n+1), & \text{if } n > 0. \end{cases}$$

Then

$$p_1(z_0) = \frac{2\Gamma(\frac{n+1}{2})}{\Gamma(\frac{n}{2})\sqrt{\pi}} \left[ -\frac{1}{n} \cos^n \theta \times {}_2F_1\left(\frac{n}{2}, \frac{1}{2}, \frac{n+1}{2}, \cos^2(\theta)\right) \right]_{\theta=\theta_0}^{\pi/2}$$

$$= \frac{2\Gamma(\frac{n+1}{2})}{\Gamma(\frac{n}{2})\sqrt{\pi}} \frac{1}{n}(1-z_0^2)^{\frac{n}{2}} \times {}_2F_1\left(\frac{n}{2}, \frac{1}{2}, \frac{n+1}{2}, 1-z_0^2\right).$$

Let $z_0 = 0$, we know that $p_1(z_0) = 1$ by the definition of probability, hence

$$\frac{2\Gamma(\frac{n+1}{2})}{\Gamma(\frac{n}{2})\sqrt{\pi}} \frac{1}{n} \sum_{k=0}^{\infty} \frac{(\frac{n}{2})(\frac{1}{2})}{(\frac{n+1}{2})k!} = 1.$$

Taking the small residuals to the right hand side, we get

$$\frac{2\Gamma(\frac{n+1}{2})}{\Gamma(\frac{n}{2})\sqrt{\pi}} \frac{1}{n} \sum_{k=0}^{n} \frac{(\frac{n}{2})(\frac{1}{2})}{(\frac{n+1}{2})k!} > 1 - \sum_{k=n+1}^{\infty} \frac{1}{2k!} = 1 - \frac{e}{2(n+1)!}.$$

Then we get a lower bound of $p_1(z_0)$

$$p_1(z_0) \geq \frac{2\Gamma(\frac{n+1}{2})}{\Gamma(\frac{n}{2})\sqrt{\pi}} \frac{1}{n}(1-z_0^2)^{\frac{n}{2}} \sum_{k=0}^{n} \left[ \frac{(\frac{n}{2})(\frac{1}{2})}{(\frac{n+1}{2})k!}(1-z_0^2)^k \right]$$

$$\geq \frac{2\Gamma(\frac{n+1}{2})}{\Gamma(\frac{n}{2})\sqrt{\pi}} \frac{1}{n}(1-z_0^2)^{\frac{n}{2}}(1-z_0^2)^n \sum_{k=0}^{n} \left[ \frac{(\frac{n}{2})(\frac{1}{2})}{(\frac{n+1}{2})k!} \right]$$

$$\geq \left(1 - \frac{e}{2(n+1)!}\right)(1-z_0^2)^{\frac{3n}{2}}.$$

This gives the statement in Lemma F.3.2. $\qquad\square$

## Footnotes

[1] It need not be unique, for now we just use them to denote any optimal solution.

[2]We did provide proof for some cases where incoherence following our new definition is significantly smaller.

[3]In [9], they use $\alpha_z = 800$, but we find it doesn't work out in our case. We will describe the difference to their experiments on Hopkins155 separately later.

[4]It makes no difference because $VV^T$ is a symmetric matrix

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

# Table of Symbols and Notations

Table 1: Summary of Symbols and Notations

| | |
|---|---|
| $\lvert \cdot \rvert$ | Either absolute value or cardinality. |
| $\lVert \cdot \rVert$ | 2-norm of vector/spectral norm of matrix. |
| $\lVert \cdot \rVert_1$ | 1-norm of a vector or vectorized matrix. |
| $\lVert \cdot \rVert_*$ | Nuclear norm/Trace norm of a matrix. |
| $\lVert \cdot \rVert_F$ | Frobenious norm of a matrix. |
| $\mathcal{S}_\ell$ for $\ell = 1, .., L$ | The $L$ subspaces of interest. |
| $n, d_\ell$ | Ambient dimension, dimension of $\mathcal{S}_\ell$. |
| $X^{(\ell)}$ | $n \times N_\ell$ matrix collecting all points from $\mathcal{S}_\ell$. |
| $X$ | $n \times N$ data matrix, containing all $X^{(\ell)}$. |
| $C$ | $N \times N$ Representation matrix $X = XC$. In some context, it may also denote an absolute constant. |
| $\lambda$ | Tradeoff parameter betwenn 1-norm and nuclear norm. |
| $A, B$ | Generic notation of some matrix. |
| $\Lambda_1, \Lambda_2, \Lambda_3$ | Dual variables corresponding to the three constraints in (A.1). |
| $\nu, \nu_i, \nu_i^{(\ell)}$ | Columns of a dual matrix. |
| $\Lambda^*, \nu_i^*$ | Central dual variables defined in Definition 2. |
| $V(X), \{V(X)\}$ | Normalized dual direction matrix, and the set of all $V(X)$ (Definition 2). |
| $V^{(\ell)}$ | An instance of normalized dual direction matrix $V(X^{(\ell)})$. |
| $v_i, v_i^{(\ell)}$ | Volumns of the dual direction matrices |
| $\mu, \mu(X^{(\ell)})$ | Incoherence parameters in Definition 3 |
| $\sigma_d, \sigma_d(A)$ | $d^{th}$ singular value (of a matrix $A$). |
| $X_{-k}^{(\ell)}$ | $X^{(\ell)}$ with $k^{th}$ column removed. |
| $r, r(\mathrm{conv}(\pm X_{-k}^{(\ell)}))$ | Inradius (of the symmetric convex hull of $X_{-k}^{(\ell)}$). |
| $\mathrm{RelViolation}(C, \mathcal{M})$ | A soft measure of SEP/inter-class separation. |
| $\mathrm{GiniIndex}(\mathrm{vec}(C_\mathcal{M}))$ | A soft measure of sparsity/intra-class connectivity. |
| $\Omega, \tilde{\Omega}, \mathcal{M}, \mathcal{D}$ | Some set of indices $(i, j)$ in their respective context. |
| $U, \Sigma, V$ | Usually the compact SVD of a matrix, e.g., $C$. |
| $C_1^{(\ell)}, C_2^{(\ell)}$ | Primal variables in the first layer fictitious problem. |
| $\tilde{C}_1^{(\ell)}, \tilde{C}_2^{(\ell)}$ | Primal variables in the second layer fictitious problem. |
| $\Lambda_1^{(\ell)}, \Lambda_2^{(\ell)}, \Lambda_3^{(\ell)}$ | Dual variables in the first layer fictitious problem. |
| $\tilde{\Lambda}_1^{(\ell)}, \tilde{\Lambda}_2^{(\ell)}, \tilde{\Lambda}_3^{(\ell)}$ | Dual variables in the second layer fictitious problem. |
| $U^{(\ell)}, \Sigma^{(\ell)}, V^{(\ell)}$ | Compact SVD of $C^{(\ell)}$. |
| $\tilde{U}^{(\ell)}, \tilde{\Sigma}^{(\ell)}, \tilde{V}^{(\ell)}$ | Compact SVD of $\tilde{C}^{(\ell)}$. |
| $\mathrm{diag}(\cdot)/\mathrm{diag}^\perp(\cdot)$ | Selection of diagonal/off-diagonal elements. |
| $\mathrm{supp}(\cdot)$ | Support of a matrix. |
| $\mathrm{sgn}(\cdot)$ | Sign operator on a matrix. |
| $\mathrm{conv}(\cdot)$ | Convex hull operator. |
| $(\cdot)^o$ | Polar operator that takes in a set and output its polar set. |
| $\mathrm{span}(\cdot)$ | Span of a set of vectors or matrix columns. |
| $\mathrm{null}(\cdot)$ | Nullspace of a matrix. |
| $\mathcal{P}_T / \mathcal{P}_{T^\perp}$ | Projection to both column and row space of a low-rank matrix / Projection to its complement. |
| $\mathcal{P}_\mathcal{D}$ | Projection to index set $\mathcal{D}$. |
| $\mathrm{Proj}_\mathcal{S}(\cdot)$ | Projection to subspace $\mathcal{S}$. |
| $\beta_1, \beta_2$ | Tradeoff parameters for NoisyLRSSC. |
| $\mu_1, \mu_2, \mu_3$ | Numerical parameters for the ADMM algorithm. |
| $J$ | Dummy variable to formulate ADMM. |
| $K$ | Used in Take-$K$-out Independence (Definition E.3.1). |