[Reviews · NeurIPS 2013]

Submitted by Assigned_Reviewer_4

The paper proposes a low-rank sparse subspace clustering (LRSSC),
which is a combination of SSC [7] and LRR [15].
LRSSC minimizes a weighted sum of the trace-norm
and the l1-norm of the representation matrix
under the self-expressiveness and the zero diagonal constraints (Eq.(1)).
Theoretical guarantees of self-expressiveness property (SEP) were obtained
in deterministic (Theorem 1) and the random (Theorem 2) setups.
Graph connectivity of LRR (not of LRSSC) is also discussed (Proposition 1).

I think this is a nice paper with several strong points
and a few minor weak points (listed below).
Although the idea of combining the trace-norm and l1-norm
is not new (as mentioned in the paper), developing its theoretical guarantees
is a significant contribution.
Furthermore, the theoretical results not only show the advantage of the proposed LRSSC,
but also imply complementary nature of SSC and LRR.
This is useful information for researchers working on subspace clustering,
and would influence their future research.

Pros:
- The proposed LRSSC is given theoretical guarantees.
- The theoretical and experimental results imply strengths and weaknesses
of SSC and LRR, and justify the proposed combination.
These insights are also useful for researchers working on SSC and LRR.
- A new definition (minimax subspace incoherence property) of subspace incoherence
is first introduced, which led to tighter bounds for SSC than the previously obtained
bounds in [20].
This new definition might be used by other theoreticians for developing better
guarantees of related methods.

Cons:
- The idea to combine the trace and the l1 norms is not new.
- The relation between the obtained theoretical bounds and the experimental
results is not clear (see the first one in 'Other comments' below).

Quality:
The paper is technically sound and the claims are supported by theory and experiment.

Clarity:
The paper is clearly written, and well-organized.

Originality:
Although the combination of the l1-norm and the trace norm has been used
in a different context [19] (as mentioned in the paper),
developing theoretical guarantees is a significant contribution.
The differences from the previous work are clearly stated.

Significance:
The paper provides important information. It theoretically and experimentally
shows the complementary nature of SSC and LRR, and justifies the proposed combination.
The information revealed in this paper on pros and cons of SSC and LRR would influence
researchers working on subspace clustering when they use existing methods,
develop new approaches, and pursue deeper understanding of what sparsity and
low-rankness of the representation matrix mean for subspace clustering.

Other comments:
- It is not clear if the experiments in Section 6.1 supports Theorem 2.
It is preferable that the range in which Theorem 2 guarantees SEP is shown
in Figure 1 if possible. If the whole experimental condition is out of
the guaranteed range, discussing how loose the guarantees would be informative.
- Since LRSSC with lambda->infinity with the zero diagonal constraint is not the same as
LRR, it is informative to show the result with LRR without the zero diagonal constraint
in Figure 1. If it is indistinguishable with LRSSC with lambda->infinity, please just
say so.
- In Remark 2, I do not really agree that 'this seems to imply that the independent subspace assumption used in [15, 16] to establish sufficient conditions for LRR to
work is unavoidable', since it has not been shown that SEP cannot be achieved with LRR unless the subspaces are independent. If the authors perhaps got some insights in the proof as said in Introduction, an explanation on why the authors think so in the
main text would be helpful.

Summary: A nice paper that proposes a combination (LRSSC) of SSC and LRR,
and gives its theoretical guarantees.
The theoretical and experimental results support the usefulness of LRSSC,
and imply complementary nature of SSC and LRR,
which is useful information for researchers working on subspace clustering.



Submitted by Assigned_Reviewer_5

SUMMARY: This paper proposes a new subspace clustering algorithm called Low Rank Sparse Subspace Clustering (LRSSC) and aims to study the conditions under which it is guaranteed to produce a correct clustering. The correctness is defined in terms of two properties. The self-expressiveness property (SEP) captures whether a data point is expressed as a linear combination of other points in the same subspace. The graph connectivity property (GCP) captures whether the points in one subspace form a connected component of the graph formed by all the data points. The LRSSC algorithm builds on two existing subspace clustering algorithms, SSC and LRR, which have complementary properties. The solution of LRR is guaranteed to satisfy the SEP under the strong assumption of independent subspaces and the GCP under weak assumptions (shown in this paper). On the other hand, the solution of SSC is guaranteed to satisfy the SEP under milder conditions, even with noisy data or data corrupted with outliers, but the solution of SSC need not satisfy the GCP. This paper combines the objective functions of both methods with the hope of obtaining a method that satisfies both SEP and GEP for some range of values of the relative weight between the two objective functions. Theorem 1 derives conditions under which LRSSC satisfies SEP in the deterministic case. These conditions are natural generalizations of existing conditions for SSC. But they are actually weaker than existing conditions. Theorem 2 derives conditions under which LRSSC satisfies SEP in the random case (data drawn at random from randomly drawn subspaces). Overall, it is shown that when the weight of the SSC term is large enough and the ratio of the data dimension to the subspace dimension grows with the log of the number of points, then LRSSC is guaranteed to satisfy SEP with high probability. I say high, because it does not tend to 1. Finally, Proposition 1 and Lemma 4 show that LRR satisfies GCP (presumably almost surely). Experiments support that for a range of the SSC weight, LRSCC works. Additional experiments on model selection show the usefulness of the analysis.

QUALITY: This is an excellent paper. It addresses a difficult and important problem. It derives new conditions that advance the state of the art for the subspace clustering problem. For the first time, easy-to-verify conditions for the case of non-independent subspaces are presented. The conditions are clean, intuitive and useful. Moreover, for the first time an attempt to study both the SEP and the GCP properties together is done. While the story is not complete yet, this is a clear step in the right direction. That said, I do have some comments to improve the paper. Most notably, the discussion in Section 4 needs improvement. Also, there are a number of imprecisions in the literature review that need to be corrected. Finally, the experiments on Hopkins 155 are not entirely clear. My detailed comments are as follows:

- For data drawn from a single subspace, I was expecting that graph connectivity would be defined as second eigenvalue of the Laplacian being larger than zero, where the Laplacian = Degree - Affinity, and Affinity = |C|+|C|^T. Somehow, the authors limit themselves to showing that all the entries of C are nonzero. It is probably trivial, but I would have preferred making the connection with connectedness (based on the eigenvalue of the Laplacian) more explicit.

- The whole paper is predicated on the idea that as one varies lambda from 0 to infinity one goes from LRR to SSC. But this is not the case, since for lambda = 0, the constraint diag(C) = 0 is not present in standard LRR. Moreover, I don't think there is a value of lambda for which one gets LRR. I was disappointed that this is not mentioned explicitly till the very end of section 4. Can one write the constraint as lambda diag(C) = 0 and the analysis follows?

- Following on the point above, it was disappointing that no result on the graph connectivity is shown for lambda > 0. In other words, the paper does not shed light on graph connectivity for SSC or LRSSC. Only for LRR.

- The Low Rank Subspace Clustering (LRSC) work of Favaro deserves a discussion. Equation in (6) is in fact an extension of LRSC, not of LRR, because LRSC uses the Frobeniuos norm of the error, while LRR uses the L21 norm. Of course they coincide in the noiseless case. Arguably, the name LRSC is more appropriate than LRR throughout, since we are not talking about a low rank representation in general, but one specifically designed for the subspace clustering problem.

- The proof of the success of LRR for independent subspaces was not shown in Vidal et al. PAMI 2005 [23]. It was shown in Section 2.3 of Vidal et al. IJCV 2008, " Multiframe motion segmentation with missing data using PowerFactorization and GPCA"

- To the best of my knowledge, the first application of subspace clustering to hybrid system identification is Vidal et al. CDC 2003, which is much earlier than Bako, Automatica 2011 [2].

- The experiments on Figure 9 seem to be showing a performance that is worse than that of SSC and LRR. As lambda goes to infinity, this should approximately converge to the SSC performance, which is 1.52% for 2 motions, 4.40% for 3 motions, and 2.18% for two motions. The reported number of nearly 9% for 3 motions seems way off. I understand that the parameter setting may be slightly different. But I wonder what happens if one uses the parameter settings of Elhamifar. Would the behavior of getting a smaller error for some lambda still hold in practice? Overall, I think the experiments needs to be redone to be made comparable with the state of the art.

- The quality of English decreases as a function of the page number.
-- The articles "the" and "a" are missing in several places (100,123,153,154,205,271,281)
-- It should be "a d-dimensional subspace" (085),
-- There should be a space before references (040,041,048,221,353),
-- It should be "these subspaces, we say a matrix C obeys the SEP" in line 100,
-- What is the \perp symbol in 115.
-- it should be "the only difference being the definition of mu" in 179
-- It should be "Assume random sampling. If ..." in 207
-- It should be "Combining Lemmas 1 and 2, we get" in line 229
-- It should be "and random sampling. It holds" in 240
-- What is probabilistic theorem? in line 245
-- It should be "ignoring constant terms" or "ignoring the constant C" or something like that in 256
-- The grammar of "The graph connectivity problem concerns when SEP is satisfied, whether each block of the solution C to LRSSC represents a connected graph" needs to be revised.
-- It should be "If the subspaces are independent, X is not full rank, and the data points" in 274
-- It should be "et al." not "et.al"
-- There is a "the the" and "if if" somewhere in the paper

CLARITY: The paper very clearly conveys the essence, originality and novelty of the contribution. But I had read the previous papers on the subject, and in particular the papers from Candes with the conditions for the deterministic case (inradius), etc. I think that a reader unfamiliar with that paper may have difficulties with definition 3 and the examples that follow. Now, the technical details inside the proofs are of course harder to follow. One minor thing is that the range of lambda values or lambda above a threshold is mentioned earlier in the paper (e.g., 182), but such a range does not appear till Remark 4, so the reader is left wondering what threshold is the paper referring to. Another issue is that it is not clear what "vice versa" refers to in Remark 4. Are you trying to say when lambda is smaller than a threshold versus larger than a threshold?

ORIGINALITY: The paper is highly original. The core contributions are to find weaker conditions for the correctness of SSC, new conditions for the correctness of LRR, and conditions for the correctness of a new method LRSSC. To the best of my knowledge, all the material presented in the paper is new.

IMPACT: Subspace clustering is receiving increased attention in the community, and sparse and low rank methods are showing excellent performance in many datasets. Understanding why these methods work is in my view fundamental. This paper presents a fundamental step towards achieving this understanding.
Summary: This paper proposes a new subspace clustering algorithm called Low Rank Sparse Subspace Clustering (LRSSC) and aims to study the conditions under which it is guaranteed to produce a correct clustering. The presented conditions are novel. This paper presents a fundamental step towards understanding why sparse and low-rank methods are appropriate for subspace clustering.

Submitted by Assigned_Reviewer_7

This paper proposes an algorithm for subspace clustering that combines nuclear-norm and L1 minimization. Theoretical guarantees show that this algorithm can succeed when the data satisfy incoherence assumptions that are weaker than in previous work. The algorithm was implemented using the ADMM method, and run on large data sets. Numerical experiments show that the algorithm can be tuned to trade off between the competing goals of separating points in different subspaces while maintaining graph connectivity between points in the same subspace.

Quality: The results are interesting and the proofs are quite substantial. The proofs use the same convex duality approach as in previous work on sparse subspace clustering (ref. 20), but are more sophisticated due to the use of nuclear-norm minimization (reminiscent of work on matrix completion).

Clarity: The paper is fairly clear, but some notation is not adequately explained: in line 91, note that |C| is the elementwise (not operator) absolute value; in line 113, the notation \set{\Lambda_1(X)} is very awkward; in line 114, the L_infinity norm and \diag^\perp notation need to be explained; in line 116, \span(X) should be the column space of X; in line 225, what is the parameter d?

Also, the supplementary material has a number of confusing points: in line 84, it would be helpful to define \supp(C), for readers not familiar with results of this type; in line 147, one should state the particular choice of \tilde{\Omega} that is needed here, and also fix the grammar in that sentence; in line 244, it is not clear what the "entry-wise box constraint \lambda" refers to, though I suspect it is constraint (5) in Lemma A.1.2; in sections A.3 and A.4, it is quite difficult to follow the logical structure of the argument, and better explanation is needed.

Originality: This work seems fairly original, in particular the notion of "minimax subspace incoherence," and the use of a singular value condition to quantify how well each subspace is represented by the observed data points.

Significance: This work seems quite significant. The proposed algorithm is especially interesting due to the trade-offs between the two goals of "separation" and "graph connectivity."
Summary: This paper has interesting results on a problem that is technically difficult, both in theory and in practice.
Author Feedback

Author rebuttal: First of all, we thank the reviewers for the high-quality comments and thoughtful input. We will revise the paper on notaion, grammar, presentation and etc. Also, we will include the references we missed out and provide short discussions on those particularly relevant.We address other comments briefly below.

To Reviewer 4:

1. About how the experiment in Figure 1 supports Theorem 2: The results shown in Figure 1 shows clearly that SEP holds for all lambda > some threshold, which is also what is predicted in Theorem 2. Though the result is only a sufficient condition, the experimental threshold is of the same order of what is predicted by Theorem 2. We will add the predicted value in the plots to further illustrate this.

2. About Figure 1 for LRR without the diag(C)=0: The experiments that produce Figure 1 is for disjoint subspaces that has total dimension exceeding the ambient dimension. In such a case, LRR without the diag(C)=0 constraint will return a trivial solution C=identity.


3. About our "seems to imply" statement of LRR in Remark 2: This is not a statement with formal proof, but we do have something in the proof of Theorem 1 suggesting that LRR requires independent assumption except very specific instances (we believe it is possible to formally show this for random generated data, but it seems out of the scope of the paper). Moreover, the synthetic data experiments support the conjecture.


To Reviewer 5:

4. About the definition for second eigenvalue of the Laplacian: Full connectivity implies the graph is a single connected body. We think this definition is more basic than the equivalent definition on the eigenvalue of the Lapalcian. In addition, the second eigenvalue of the Laplacian is difficult to analyze.


5. About lambda diag(C)=0 constraint: This is a good idea, but for the analysis, lambda->0 does not reduce to LRR without the diagonal constraints, unless lambda actually reaches 0. In fact, we think the constraint on the diagonal entries is natural in the problem of subspace clustering, since representing one data point fully or partially with itself is not meaningful in constructing the affinity graph. Without this constraint, the solution of LRSSC is trivial (identity) in disjoint subspace settings (e.g., that in Figure 1).


6. About our connectivity result: It is a very preliminary (and simple) result. We understand the disappointment that it is for LRR only, but not LRSSC. We hope to address this and obtain stronger and more general graph connectivity result for subspace clustering in a future work.

7. About the experiments on Hopkins155: We will study Elhamifar's code, redo the experiment and try to make the results comparable. The results on LRR end being worse than LRR is however understandable since for fair comparison we did not use any post processing of the affinity graph.

8. About the clarity in Remark 4: The threshold in Remark 4 is different from else where in the paper. In Remark 4, the threshold refers to the range when (2) is stronger than (3). The "threshold" in other parts of the paper refers to the range of lambda that results in a solution satisfying SEP. This can be seen from either (2) or (3). We will make the statement clearer.

To Reviewer 7:

9. Line A.244: Yes, the entry-wise box constraint refers to (5) in Lemma A.1.2. We will make the statement clearer.

10. Line 255: d is the dimension of each subspace.